# Climatic and anthropogenic drivers of a drying Himalayan river

Gopal Penny[1,2,3], Zubair A Dar[4], and Marc F Müller[1,2]

[1]Environmental Change Initiative, University of Notre Dame, Indiana, USA
[2]Civil and Environmental Engineering and Earth Sciences, University of Notre Dame, Indiana, USA
[3]School for Environment and Sustainability, University of Michigan, Ann Arbor, Michigan, USA
[4]Energy and Resources Group, University of California, Berkeley, California, USA

**Correspondence:** Gopal Penny (gopalpenny@gmail.com)

**Abstract.** Streamflow regimes are rapidly changing in many regions of the world. Attribution of these changes to specific hydrological processes and their underlying climatic and anthropogenic drivers is essential to formulate effective water policy. Traditional approaches to hydrologic attribution rely on the ability to infer hydrological processes through the development of catchment-scale hydrological models. However, such approaches are challenging to implement in practice due to limitations in using models to accurately associate changes in observed outcomes with corresponding drivers. Here we present an alternative approach that leverages the method of multiple hypotheses to attribute changes in streamflow in the Upper Jhelum watershed, an important tributary headwaters of the Indus basin, where a dramatic decline in streamflow since 2000 has yet to be adequately attributed to its corresponding drivers. We generate and empirically evaluate a series of alternative and complementary hypotheses concerning distinct components of the water balance. This process allows a holistic understanding of watershed-scale processes to be developed, even though the catchment-scale water balance remains open. Using remote sensing and secondary data, we explore changes in climate, surface water, and groundwater. The evidence reveals that climate, rather than land use, had a considerably stronger influence on reductions in streamflow, both through reduced precipitation and increased evapotranspiration. Baseflow analyses suggest different mechanisms affecting streamflow decline in upstream and downstream regions, respectively. These findings offer promising avenues for future research in the Upper Jhelum, and an alternative approach to hydrological attribution in data-scarce regions.

## 1 Introduction

Water resources are changing throughout the world under anthropogenic pressures including climate change, land use change, and changes in water management (Vörösmarty et al., 2004; Milly et al., 2008; Ceola et al., 2019). These drivers pose challenges for water policy by increasing climatic variability (Smirnov et al., 2016), exacerbating water scarcity (Srinivasan et al., 2017), and reducing our ability to predict hydrological variables (Ehret et al., 2014). In arid and semi-arid regions, these concerns are particularly alarming given the concurrent challenges of increasing hydrological uncertainty and competition for scarce water resources (Flörke et al., 2018; Aeschbach-Hertig and Gleeson, 2012). In many such regions, mitigation and adaptation strategies are urgently needed but require accurate understanding of the underlying drivers of change (Thompson et al., 2013; Penny et al., 2020a). In order to address the management challenges of mitigation and adaptation, observed changes

in hydrological processes must be correctly attributed to the corresponding drivers (i.e, the processes that cause changes in hydrology).

Here, we focus on attribution of hydrological changes in the Upper Jhelum watershed, a headwater catchment of the Indus basin and an important source of water for both India and Pakistan (Romshoo, 2012). The Upper Jhelum provides essential ecosystem services in the Kashmir valley, yet many of these services have been threatened in recent decades with lakes and wetland shrinking, fish populations declining, and less water available for agriculture (Wetlands International South Asia, 2007). Each of these ecosystem services depends on streamflow, which has declined dramatically in the Upper Jhelum since the year 2000. This decline is concerning given that the watershed is increasingly vulnerable to water stess due to population growth (Showqi et al., 2014), increasing pollution (Rather et al., 2016), and climate change (Rashid et al., 2015). These concerns are compounded by the transboundary nature of the Upper Jhelum watershed, and yet attribution is complicated by the myriad of potential drivers and changing processes occurring within the basin. Streamflow is fed by a variety of sources within the basin, including precipitation, seasonal snowmelt, and glacier melt (Romshoo et al., 2015). The valley contains a thick surficial aquifer which is recharged directly from the surrounding karst mountains, meaning that there may be considerable flows that bypass tributary streams (Jeelani, 2008). Furthermore, the large scale of the basin (nearly 13 000 km$^2$), varying topography (elevation spans 1500 m to over 5000 m above sea level), and rapidly changing land use (agricultural intensification and increasing orchards, see Rather et al., 2016) make it such that adequately estimating hydrological variables is difficult.

This research builds upon previous studies that associated declining streamflow in the Upper Jhelum with potential drivers of change. For instance, Romshoo et al. (2015) found that declining streamflow in the headwaters of the Upper Jhelum were associated with glacier recession and changes in snowmelt in the basin. Another study by Romshoo and Ali (2018) identified negative trends in precipitation as a key driver of losses in streamflow, and analysis by Zaz et al. (2019) indicated that changes in precipitation may have resulted from global warming. Badar et al. (2013a) identified changing land use change as a key contributor to changes in runoff, but the implications on streamflow changes in the river remained unclear. Considerable hydrological research at the basin scale has focused on streamflow forecasting (e.g., Mahmood and Jia, 2016; Badar et al., 2013b), along with empirical work characterizing streamflow in tributaries and the hinterlands (Jeelani, 2008; Jeelani et al., 2013; Romshoo et al., 2015). These studies have tended to focus on particular aspects of hydrological change and a coherent understanding of the causes of hydrological change has therefore yet to be achieved.

Given the importance of this river, scientific understanding of changing hydrological processes is essential to support effective domestic and transboundary water management. Nevertheless, the difficulties in conducting hydrological attribution in this basin are common to many regions, as measurement challenges often make it difficult to accurately close the water balance (Kampf et al., 2020) and changes in its key components may go unnoticed. These difficulties make the broader challenge of attributing hydrological changes *ex post* to their landscape and climatic drivers a substantial ongoing scientific challenge (Wine and Davison, 2019).

The most common approaches to hydrological attribution (see Dey and Mishra, 2017; Luan et al., 2021, for recent reviews) are unlikely to be appropriate for the Upper Jhelum. In perhaps the most common approach, watershed modeling is used to simulate streamflow, and the calibrated model is used to infer hydrological relationships and conduct the attribution (e.g., Liu

et al., 2019). Goodness-of-fit and other model evaluation metrics influence which models or calibration parameters are given more credibility (Müller and Thompson, 2019) and which models are, in turn, used to identify the causal processes in the attribution. The major challenge of this approach for attribution is the difficulty in validating hydrological processes within the model. These difficulties can arise due to equifinality (cannot distinguish drivers by considering outcomes only, Beven, 2006), nonlinearity (drivers are not linearly separable, Sivapalan, 2006), hydrological regime shifts (Foufoula-Georgiou et al., 2015; Gober et al., 2017), or lack of appropriate data (cannot test hypotheses pertaining to specific processes, Sheffield et al., 2018). Calibrated models have particular difficulty with hydrological regime shifts, where the underlying mechanisms of streamflow generation change but cannot be directly observed (Savenije, 2009). This general approach to attribution is known as predictive inference (Ferraro et al., 2019) because the accuracy of predictions is used to validate the model which is then used to infer the underlying causal processes. Another approach to attribution utilizes hydrological fingerprinting, but this approach requires that the effects of specific drivers (i.e., "signatures") are specified *a priori* and used to identify or separate the effects of multiple drivers in particular situations (Viglione et al., 2016). This approach is not suitable to analyzing streamflow decline the Upper Jhelum because multiple drivers may have similar signatures in terms of streamflow outcomes. A third set of approaches use the relationship between the water balance and energy balance (via the latent heat flux) to assess how precipitation is partitioned into streamflow and evapotranspiration as a basis for understanding changes over time (e.g., Ning et al., 2018; Tomer and Schilling, 2009). Although these approaches associate changes in streamflow with changes in climate or land use, the partitioning approach is more akin to correlative than than causal analysis. Lastly, field-based studies, including paired catchment studies, have also been used for hydrological attribution (Penny et al., 2020b), but may require extensive resources and often cannot be applied in situations of historical attribution, unless a suitable space-for-time substitution can be applied. Additionally, field research may be logistically complicated where accessibility is challenging (e.g., due to remoteness or political instability) and considerable methodological challenges remain, especially in human-impacted catchments, due to the difficulty in capturing spatial complexity and ruling out potential alternative drivers of change (Srinivasan et al., 2015).

Here, we advance an alternative approach to attribution, wherein several plausible hypotheses are generated and each is evaluated *separately*. The approach is grounded in the method of multiple working hypotheses (Chamberlin, 1965) and has two key benefits. First, the approach does not rely on constructing and calibrating a fully integrated catchment-scale hydrological model, but rather relies on specific sets of empirical observations that are necessary to individually test each hypothesis. As such, it is particularly helpful in data-scarce catchments, where limited hydrological records preclude complete characterization of the water balance and data collection may be difficult. Relatedly, the approach allows us to explicitly account for uncertainties in hydrological fluxes or issues with data integrity. These issues might cause some (though not all) hypotheses to be inconclusively evaluated. This contrasts with the predictive inference approach, where integrity issues with some of the data might cause the whole analysis to be inconclusive, or worse, be concealed within the model calibration process (e.g., due to equifinality issues). Second, the approach mitigates against observational and conceptual biases in which certain hypotheses are favored based on preconceived notions of change (Railsback et al., 1990). Using consistent observational datasets before and after 2000 ensures that only a nonstationary bias would affect the results of the analyses. Further, by employing separate analyses for each hypothesis, we construct a process-based narrative of attribution in which each analysis comprises part of

the whole and serves to corroborate or contradict other analyses. Indeed, the method of multiple hypotheses advocates broader understanding of the whole over in-depth understanding of individual components. By favoring breadth over depth we seek to develop a coherent and holistic narrative of change.

Although initially proposed in 1890 (and later republished as Chamberlin, 1965), few studies have applied the method of multiple hypotheses towards hydrological attribution. Harrigan et al. (2014) used the method of multiple hypotheses to demonstrate the combined effect of changing precipitation and catchment drainage in producing greater streamflow. Srinivasan et al. (2015) sought to attribute hydrological change in a drying river by generating (and subsequently testing) a set of hypotheses based on stakeholder knowledge. Here, we employ the water balance as a guiding framework to generate hypotheses regarding hydrological processes while using stakeholder knowledge via published literature to provide additional context with respect to water management.

Previous studies using the method of multiple hypothesis to attribute hydrological change were grounded in building empirical evidence through extensive field research. However, given the challenges associated with field research (described above), remote sensing offers tremendous potential to overcome both major limitations of field experimentation in terms of capturing spatial heterogeneity and generating observational datasets *after* hydrological changes occur and are detected (e.g., Valentín and Müller, 2020). The increasing availability of remote sensing products now provides satellite estimates of precipitation, evapotranspiration, and changes in water storage including surface water, soil moisture, and groundwater (Montanari and Sideris, 2018). The Landsat record provides coverage extending back to 1972, and many relevant satellites were launched circa 2000, providing a record over the last two decades. Remote sensing therefore provides unique opportunities to attribute hydrological change in data-scarce regions (Müller et al., 2016; Penny et al., 2018), particularly when combined with secondary data (i.e., historical data collected by third parties, such as government agencies). This study demonstrates this potential by using secondary data and remote sensing observations to apply the method of multiple hypotheses in the Upper Jhelum watershed.

## 2 Study site: The Upper Jhelum watershed

The Upper Jhelum Watershed in the Western Himalayas lies between the Karakorum mountain range in the north, the Pir Panjal range in the south and west and Zanskar range in the east. The river serves as one of the six main tributaries of the Indus (Fig. 1a), supporting residents in both India and Pakistan. The watershed covers approximately 13,000 km$^2$ and drains into the Kashmir valley where the river flows east to west before discharging through a deep gorge at the outlet along the western side of the basin (Fig. 1b). The valley rests upon a layer of alluvial sediment which partially overlays a layer of glacial till, the combination of which extends nearly 1 km into the subsurface (Dar et al., 2014). Mountain ranges on either side of the valley consist of limestone Karst formations, and streamflow is supported both by groundwater recharge in the valley and karst springs emerging from mountainsides (Jeelani, 2008).

The climate in Kashmir is characterized by four seasons (Mahmood et al., 2015) including winter (December–February), spring (March–May), summer (June–August), and autumn (September–November). The valley receives an average annual precipitation of 700–1250 mm per year, depending on elevation. Although precipitation occurs throughout the year, it is

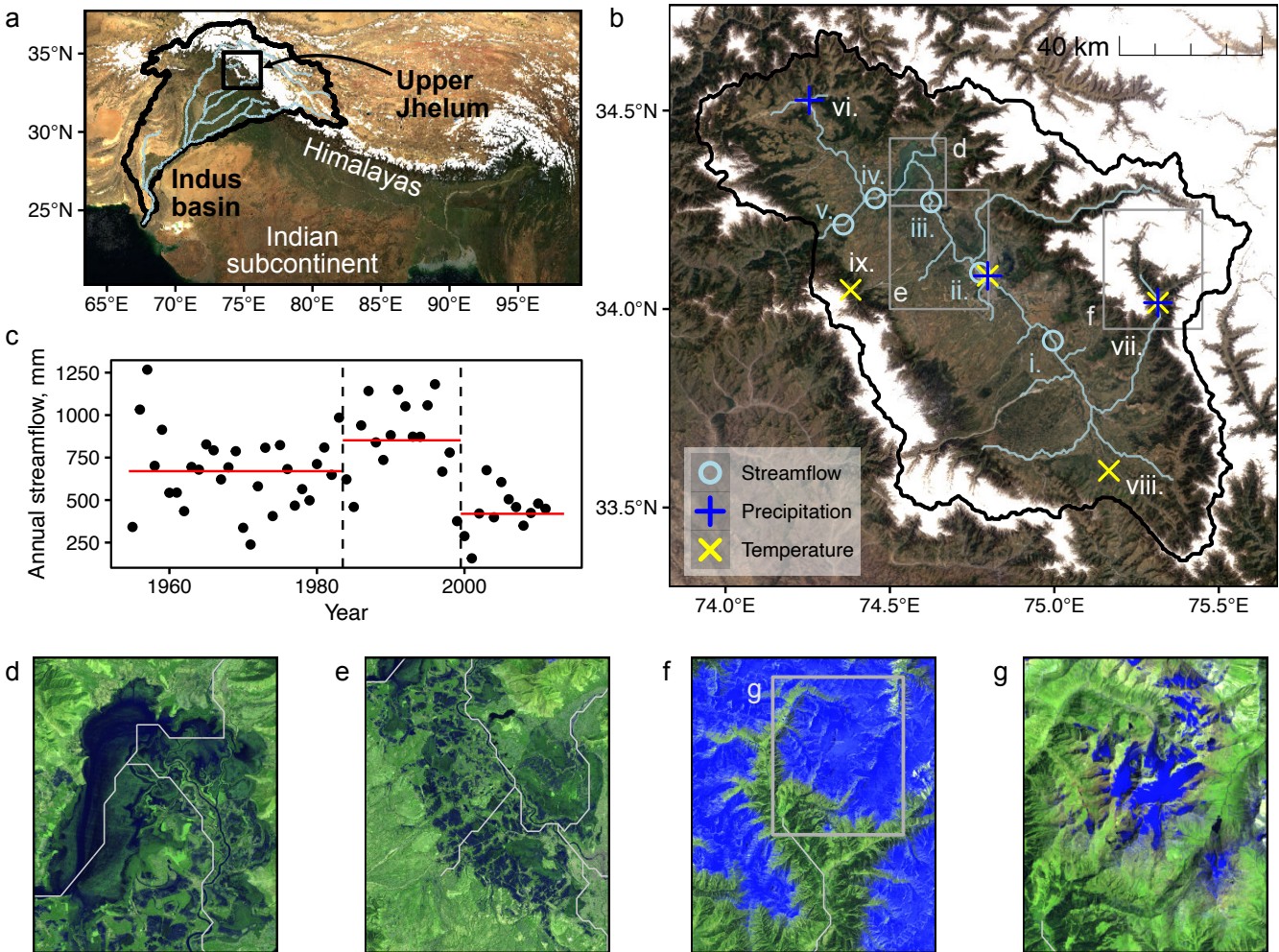

**Figure 1.** Study site and hydrological change. (a) Himalayan water towers including the Indus basin and location of the Upper Jhelum. (b) Upper Jhelum watershed including locations of streamflow records (gauges i-v), climate records (stations ii and vi-ix), and bounding boxes for (d-f). The streamflow gauge numbers refer to (i) Sangam, (ii) Munshi Bagh, (iii) Asham, (iv) Sopore, and (v) Baramulla. (c) Annual streamflow normalized by catchment area, with long-term averages for the periods 1955–1983 (670 mm, no climate data), 1984–1999 (852 mm, pre-2000 with climate data), and 2000–2013 (419 mm, post-2000). The remaining panels highlight water storage within the Upper Jhelum including (d) Wular lake, (e) valley inundation, (f) snowpack in May, and (g) Kolahoi glacier in August. In these images (d-g), Landsat bands swir2-swir1-red are mapped to red-green-blue, so that vegetation, water, and snow are clearly visible as green, dark, and blue pixels, respectively. MODIS imagery in (a) (Vermote, 2015) and Landsat composite imagery in (b, d-g) from U.S. Geological Survey were prepared and downloaded using Google Earth Engine (Gorelick et al., 2017).

dominated by Mediterranean westerlies in the spring (March–May) and the Indian monsoon in the summer (June–September)

(Zaz et al., 2019). Average valley temperatures range from $8°C$ in January to $29°C$ in August. Higher elevations remain below freezing in winter and streamflow receives a boost from snowmelt as temperatures warm throughout the spring.

The main stem of the upper Jhelum is intercepted by Wular lake (Fig. 1d) between gauges (iii) and (iv). A number of other, smaller lakes intersect tributaries within the watershed and seasonally inundated wetlands occupy much of the center of the valley (Fig. 1e). Tributaries connect the valley to surrounding hill stations where tourism services are a mainstay of economic production (Malik and Bhat, 2015). The Upper Jhelum also provides transport services, for ferrying people as well as timber extracted from forests (Raina, 2002). Agricultural land comprises the majority of the valley and supports paddy, maize, and wheat, as well as noticeably increasing fruit orchards (DES, 2015). Summer is the primary growing season for paddy and maize, which are sown in the spring and harvested in the late summer or early autumn. Wheat is typically planted in October and harvested the following June. Outside of the valley bottom, the Upper Jhelum watershed consists primarily of grassland, shrubs, and coniferous forests (Alam et al., 2020).

## 2.1 Declining streamflow

Streamflow observations were obtained from the The Department of Irrigation and Flood Control for Jammu and Kashmir at five stream gauging stations throughout the Upper Jhelum catchment (Fig. 1b) for the period 1955-2013. Standard data integrity checks (Searcy and Hardison, 1960) were applied to ensure temporal and spatial consistency (see Supplementary Information, Sect. S1). Streamflow in the main stem of the Upper Jhelum has declined over time. In particular, annual streamflow timeseries reveal a dramatic decline around the year 2000 (Fig. 1c). Average annual streamflow at the watershed outlet at Baramulla station (Fig. 1b, gauge *v*) during the 2000-2013 period ($419 \mathrm{\ mm \ y^{-1}}$) reduced by 50% compared to the 1984-1999 period ($852 \mathrm{\ mm \ y^{-1}}$), and by 27% compared to the 1944-1983 baseline ($670 \mathrm{\ mm \ y^{-1}}$). Similar declines were observed at the four additional streamflow gauges (see Fig. S2). Multiple hypotheses on the drivers and hydrological processes underlying this streamflow decline are introduced and evaluated in the remainder of the paper.

## 3 Methods

### 3.1 Implementing the method of multiple hypotheses

We use the water balance as a guiding framework to attribute the decrease in Jhelum streamflow. In particular, it serves as a conceptual tool to build alternative hypotheses regarding changes in each of the water balance fluxes that could explain the observed reduction in streamflow (Fig. 2). Testing these hypotheses individually allows us to build understanding about the different pathways of hydrologic change throughout the catchment. Importantly, this approach does not rely on the predictive ability of an aggregate hydrological model, where calibration would depend upon on specific information that is not necessarily available. The multiple hypotheses are developed for each component of the water balance in the following paragraphs, along with approaches and datasets (Fig. 2, Table 1) to evaluate these hypotheses.

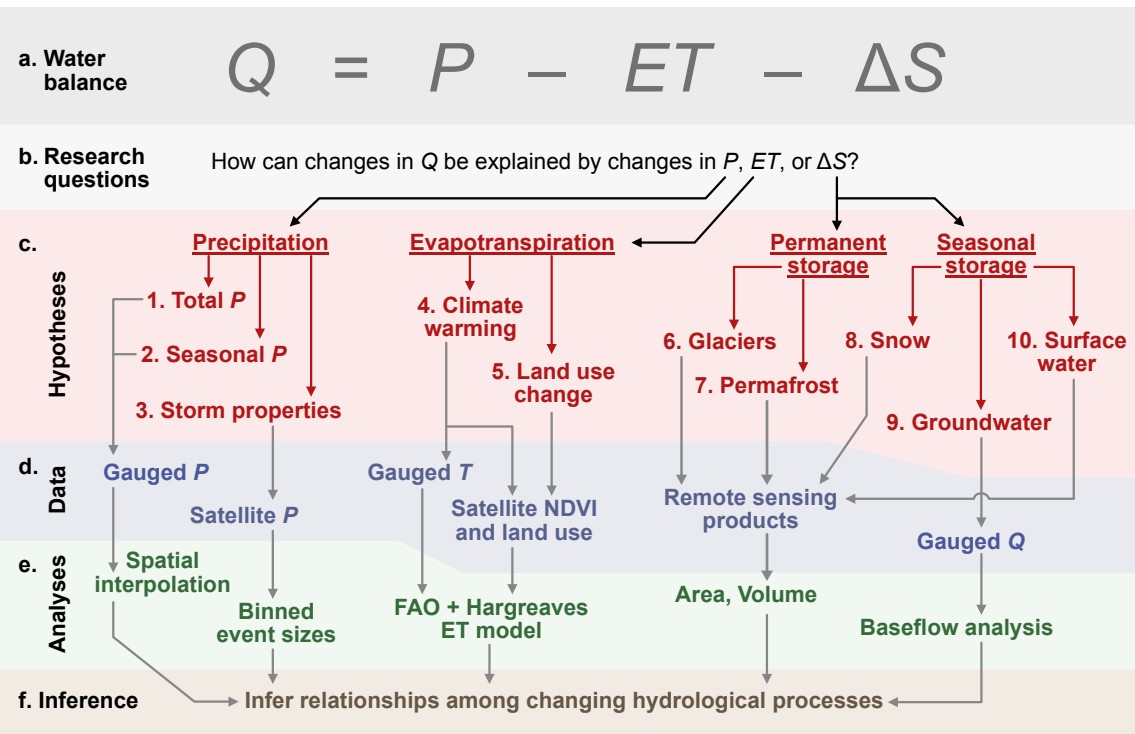

**Figure 2.** Hypothesis generation and evaluation. We develop a general empirical approach to attribution that (a) utilizes the water balance to (b) generate overarching research questions and (c) construct multiple hypotheses. The hypotheses are addressed (d) using remote sensing and *in situ* secondary data and (e) by applying empirical analyses, the results of which are (f) used to infer relationships regarding changing hydrological processes. We implement this approach in the Upper Jhelum watershed by defining the long-term water balance in terms of streamflow ($Q$), precipitation ($P$), evapotranspiration ($ET$), and changes in catchment water storage ($\Delta S$). Specific research questions (red), data (blue), and analyses (green) focus on the detection and characterization of changing water balance fluxes in the Upper Jhelum. The inference step is contingent on understanding *how* each flux has changed and provides a deeper understanding of watershed hydrological processes.

## 3.2 Precipitation

### 3.2.1 Hypotheses

Changes in precipitation ($P$) could occur through different mechanisms that would have distinct impacts on streamflow. For instance, a reduction in precipitation would decrease the water balance inputs and reduce the water available to generate streamflow. Alternatively, an increase in storm frequency and reduction in average storm size could produce an increase in vadose zone water storage, greater evapotranspiration (ET), and reduced streamflow (Zhao et al., 2019), even if aggregate precipitation volumes remain unchanged. We therefore pose the following hypotheses:

– Hypothesis 1: Annual precipitation declined.

**Table 1.** Datasets used in analyses.

| Dataset | Type | Source | Time period | Recurrence |
|---|---|---|---|---|
| Streamflow | Station | The Department of Irrigation and Flood Control (J&K Govt) | 1955–2013 | Daily–monthly |
| Precipitation | Station | Indian Meteorological Department (Srinagar) | 1984–2013 | Monthly |
| Temperature | Station | Indian Meteorological Department (Srinagar) | 1984–2013 | Monthly |
| Precipitation | Satellite | PERSIANN | 1984–2013 | Daily |
| Evapotranspiration | Satellite | MODIS ET Product | 2000–2013 | 8-daily |
| Land use | Satellite | MODIS Land cover Product | 2001, 2010 | Annual |
| Snow and water extent analyses | Satellite | Landsat Missions 5, 7, 8 | 1989–2013 | Varies |

- Hypothesis 2: Climate exhibited a change in rainfall seasonality.

The first hypothesis represents a direct reduction in the water input to the catchment that would generate a corresponding reduction in streamflow. The second hypotheses represents a shift in the availability of water throughout the year. Such a shift in precipitation from a season with low ET to high ET could decrease average streamflow.

- Hypothesis 3: The distribution of precipitation events changed.

A shift in the distribution of rainfall events could have various effects. A shift towards smaller event sizes would likely reduce the "fast" component of streamflow (i.e., quickflow, McCaig, 1983) and leave a greater fraction that is directly intercepted, re-evaporated, or infiltrated. In other words, a shift in precipitation patterns towards more frequent small events and fewer large events could lead to a reduction in runoff and increase in evapotranspiration. These changes would reduce quickflow and groundwater recharge, and thus ultimately reduce annual streamflow volume.

### 3.2.2 Data sources

To address these hypotheses, monthly precipitation data were obtained from the Indian Meteorological Department (IMD, Srinagar) for the period 1984–2013 at Srinagar, Kupwara, and Pahalgam stations (stations ii, vi, and vii in Fig. 1b). We interpolated monthly precipitation to the watershed scale using Thiessen polygons. We also conducted an interpolation using the inverse-distance-squared approach with an elevation gradient, but this method appeared to overestimate precipitation at the water balance scale (see Sect. S2, Fig. S6). Consequently, we only used Thiessen-interpolated rainfall to test our hypotheses and note that this decision did not introduce any bias in our analysis because both interpolations produced similar results in terms of the change in precipitation (Fig. S6).

The monthly frequency of the IMD precipitation dataset precluded analysis of the distribution of precipitation event sizes. Therefore, we utilized daily precipitation records from the gridded PERSIANN Climate Data Record (CDR, Ashouri et al., 2015). The purpose of PERSIANN CDR is to provide consistent precipitation data for long-term climate analysis dating back to 1984 at 0.25 degree resolution. Consistent with the daily frequency of PERSIANN rainfall data, we here define a precipitation event as any day with precipitation $\geq$ 2 mm. In order to determine a shift in the distribution of event sizes, we binned events into three groups: small (2–7.4 mm), medium (7.4–18.6 mm), and large ($\geq$18.6 mm) events. The size of the bins was determined such that the total precipitation from all events within each bin was equivalent (i.e., the sum of precipitation from all events in the small bin was equivalent to the sum in the medium and large bins). We then determined whether or not the number of precipitation events in each bin changed before and after 2000.

To formally test these hypotheses, we bootstrapped confidence intervals ($N = 10^4$) for annual precipitation, seasonal precipitation, and the number of precipitation events in each category. In each case, the null hypothesis (no change before and after 2000) was rejected if the 95% confidence interval excluded zero. As a robustness check for the change in event size, we re-ran the same analyses using minimum event sizes of 1 mm and 3 mm and adjusting the bins accordingly. The hypothesis tests resulting from these robustness checks yield identical results (Sect. S2, Table S1).

### 3.3 Evapotranspiration

#### 3.3.1 Hypotheses

Evapotranspiration (ET) affects streamflow by reducing the volume of water stored in the vadose zone that could have otherwise produced streamflow. A reduction in streamflow could therefore be generated by an increase in ET, either through changes in potential evapotranspiration or vegetation properties and land use. We therefore include the following hypotheses:

– Hypothesis 4: Climate change and warmer air temperatures led to greater evapotranspiration.

– Hypothesis 5: Land use change toward water-intensive crops led to greater evapotranspiration.

Both hypotheses are grounded in observed, ongoing changes within the watershed. Temperatures have been warming (Zaz et al., 2019) and there has been a notable shift towards orchard plantations in portions of the valley (Romshoo and Rashid, 2014), which may use more water than traditional crops due to a longer growing season (Allen et al., 1998) and better access to subsurface water storage (Zhang et al., 2018). Both changes might have led to increased evapotranspiration and a reduction in streamflow.

#### 3.3.2 Data Sources

We required an approach to estimate seasonal and annual evapotranspiration for the periods before and after 2000. Multiple approaches have been developed to estimate evapotranspiration (ET) using remote sensing products, but few provide robust estimates that would allow consistent comparisons of the the pre- and post-2000 periods. For instance, MODIS provides an 8-day ET product (Mu et al., 2013), but this dataset is only available since the launch of the Terra satellite in late 1999. The

surface energy balance approach (SEBAL, Bastiaanssen et al., 1998) can be used to estimate ET from Landsat imagery prior to 2000, but relatively few Landsat images were available during this time period. The instantaneous nature of SEBAL may therefore present biased estimates of seasonal ET, as usable Landsat imagery is predominantly available for cloudless days that may not be representative of average seasonal ET conditions.

We therefore applied a regression framework based on the concept of reference evapotranspiration (Allen et al., 1998), where actual evapotranspiration ($ET$) is defined as the product between the evapotranspiration value ($ET_0$) of a reference crop under well-watered conditions, and a crop coefficient ($k$). These two terms respectively capture the effect of climate ($ET_0$) and land use ($k$, representing the effect of the type of vegetation on its ability to evapotranspire water). For calculating reference ET, we used the Hargreaves equation (Hargreaves and Samani, 1985), which provides estimates of $ET_0$ at monthly timescales or larger.

Hargreaves equation captures the effect of climate drivers and accounts for extraterrestrial radiation ($R_a$), air temperature ($T_A$), and the diurnal temperature range ($T_R$):

$$ET_0 = 0.0023 R_a (T_A + 17.8)\sqrt{T_R}. \tag{1}$$

Extraterrestrial radiation $R_a$ varies seasonally with changing solar declination angles which are associated with latitude and topography. The diurnal temperature range $T_R$ depends on a variety of climate conditions including humidity, soil moisture, precipitation, and cloud cover (Dai et al., 1999; Geerts, 2003). We group these parameters into a single seasonal parameter, $a_s \propto$

$R_a T_R^{0.5}$, which represents mean climate conditions within each season and captures the seasonal variability in the Upper Jhelum watershed. We assume that the greenness of the vegetation canopy exerts the primary control on the stomatal conductance, such that we can define the crop coefficient as $k \approx NDVI^c$, where $c$ is a calibrated parameter and NDVI is the normalized-difference vegetation index (Carlson and Ripley, 1997). This assumption is supported by empirical evidence (Duchemin et al., 2006; Groeneveld et al., 2007). Combining the Hargreaves Equation with this parameterization of the crop coefficient $k$ allows

evapotranspiration at any pixel ($i$) season ($s$) and year ($y$) to be expressed as a nonlinear regression:

$$ET_{i,s,y} = a_s \times (T_{A,i,s} + 17.8) \times NDVI_{i,s,y}^c + \varepsilon_{i,s,y} \tag{2}$$

where $ET_{i,s,y}$ is average MODIS ET for an individual pixel $i$ in season $s$ and year $y$, $T_{A,i,s}$ is the interpolated post-2000 seasonal average air temperature (see Sect S3), and $NDVI_{i,s,y}$ is the average Landsat NDVI for the same year-season-pixel combination. The regression coefficients $a_s$ represent the estimated average effect of extra-terrestrial radiation $R_a$ and diurnal temperature range $T_R$, and $c$ mediates the effect of NDVI on the the crop coefficient. These coefficients are assumed to be

stationary across pixels (homogeneous) and years (stationary). We used a cross-validation analysis to evaluate how robust our ET estimates were to uncalibrated data. A calibration sample was formed by independently drawing 80% of pixels from each image, which we used to to estimate the regression coefficients $a_s$ and $c$. The estimated coefficients were then used to predict $ET$ on the remaining 20% of the pixels using Equation 2. Predictions matched $ET$ observation on the validation with a high degree of accuracy ($R^2 = 0.87$). The cross-validation results give confidence in the assumptions that $c$ and $a_c$ were stationary

and homogeneous. We relied on these assumptions to predict pre-2000 $ET$ using Equation 2 and regression coefficients $a_s$ and $c$ estimated using post-2000 observations.

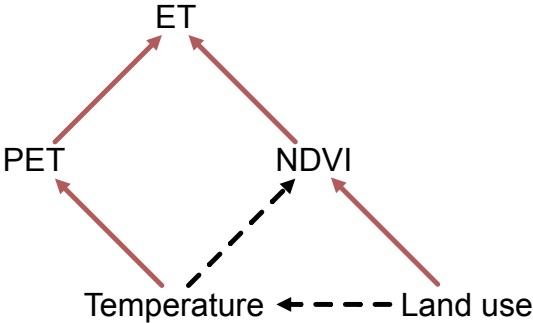

**Figure 3.** Separating local and climatic drivers of evapotranspiration (ET). Temperature (T) modulates ET directly through the effect on potential evapotranspiration (PET), or indirectly by changing growing conditions, vegetation structure and phenology, and therefore NDVI. Land use modulates ET by directly changing land cover characteristics, or indirectly by affecting air temperature. We evaluate the effects depicted by the solid red lines and decouple the effects of the dashed lines via the Hargreaves model and maps of land use change (see text for details).

Temperature and land use modulate evapotranspiration by affecting potential evapotranspiration (PET) and vegetation characteristics (e.g., morphology, stomatal conductance). Within the hypothesis testing framework, we assume that the primary control of temperature on ET occurs through the effect on PET, as described in Eq. 1, and that the effect of land use occurs primarily through the effect on NDVI. In order to account for the possibility of additional links among temperature, land use, and ET (see Figure 3), we provide additional robustness checks as described below and presented in the Results (Sect. 4).

We tested both the hypotheses that temperature (H4) and land use (H5, via NDVI) increased evapotranspiration by bootstrapping confidence intervals in each season. For temperature, treated each year-season as an independent observation to bootstrap confidence intervals ($N = 10^4$) on the change in seasonal temperature and used the results to predict the effect on evapotranspiration. For NDVI, we assumed that the uncertainty occurred due to our inability to precisely measure the relationship between pixel NDVI and ET. We therefore created NDVI bins (with width of 0.02) and boot strapped confidence intervals ($N = 10^4$) for ET by sampling the associated ET for each pixel NDVI value from the associated NDVI bin.

As a robustness check to better evaluate the above assumptions, we evaluated changes in land use based on the 17 land use categories proposed by Strahler et al. (1999) using MODIS imagery. We combined these categories into six super classes to represent major land use categories within the watershed: grassland and shrubs, forest, cropland, mosaic vegetation, open water (lakes), urban, and barren land. Within the Upper Jhelum, the mosaic vegetation class indicates the presence of orchard plantations (see Fig. S9). Land use change involving the cropland and mosaic classes are therefore likely to have an outsize effect on ET in the watershed, representing the primary component of locally-driven anthropogenic changes. We used the MODIS land cover classification from 2001 (the earliest available year) to approximate land use prior to the observed hydrological change, and the land classification from 2010 to represent land use after the change.

### 3.4 Catchment storage

#### 3.4.1 Hypotheses

The Upper Jhelum contains a variety of storage reservoirs including surface water bodies, groundwater, snow, and glaciers (Figure 1 d-g). From the water balance, a decrease in storage (e.g., glacial melt) must be matched by a corresponding increase in the outgoing fluxes, evapotranspiration or streamflow. As such, a long-term decrease in catchment storage could have produced a corresponding increase in streamflow followed by a similar decrease as catchment storage stabilizes. We hypothesize that there could have been a long-term reduction in permanent water storage leading to a subsequent decline in streamflow. For instance:

- Hypothesis 6: A long-term decline in glaciers produced an increase and subsequent decrease in streamflow.

- Hypothesis 7: A long-term decline in permafrost produced an increase and subsequent decrease in streamflow.

These hypotheses are grounded in studies of other catchments showing that a warming climate could temporarily increase streamflow through glacial loss (Singh and Kumar, 1997; Schaner et al., 2012) or permafrost melting (Kurylyk et al., 2016; Qiang et al., 2019). Such changes have been predicted to occur in high-elevation montane regions and could have contributed to the increase in streamflow in the 1980s and 1990s and subsequent decline after 2000 (Figure 1c).

In addition to these long term effects, catchment storage at the *seasonal* time scale creates a time lag between precipitation and streamflow. Understanding how snow cover, groundwater, and surface water storage have changed over time would provide additional insight into the processes governing hydrological change in the catchment. Specifically, we hypothesize that:

- Hypothesis 8: Earlier snowmelt contributed to an earlier peak in the annual hydrograph.

- Hypothesis 9: Groundwater storage in the saturated zone of the riparian aquifer declined.

- Hypothesis 10: Surface water storage in lakes and wetlands in the valley decreased.

The processes involved in these three hypotheses might have increased the proportion of annual precipitation exiting the catchment as evapotranspiration instead of streamflow. All things being equal, earlier snowmelt would tend to increase the amount of streamflow early in the year and decrease streamflow later in the year. Earlier snowmelt would also allow greater vegetation activity and evapotranspiration earlier in the spring, thereby reducing groundwater recharge. A storage reduction, both in the saturated zone of the riparian aquifer and in the riparian lakes and wetlands in the valley bottom, could be indicative of reduced seepage and increased evapotranspiration from the unsaturated vadose zone (see Discussion in Sect. 5).

#### 3.4.2 Data sources: Glaciers and permafrost

High altitude hillslopes and mountain peaks in the Upper Jhelum exhibit sufficiently cold annual temperatures to support both glaciers and permafrost. In particular, Kolahoi glacier sits along the northeastern edge of the watershed and has been melting

over recent decades. The glacier was approximately 14.5 km$^2$ in 1962 and 11.3 km$^2$ in 2014 (Shukla et al., 2017).We rely on estimates of glacial mass loss from published studies to determine whether these losses are sufficient to explain the observed variations in streamflow. In particular, we use the average annual loss of glacial mass as an upper bound on the potential for deglaciation to have contributed to the observed reduction in streamflow. Rashid et al. (2017) estimated that Kolahoi glacier lost 0.3 km$^3$ of volume over the period 1962 to 2010, but did not estimate errors associated with this value. Instead, we approximate confidence intervals using estimates of the error on areal changes in the Glacier from Shukla et al. (2017), who provided bounded estimates on the loss of glacier area over time as $3.18 \pm 0.34$ km$^2$ over the period 1962–2014. By assuming the volumetric loss was associated with the midpoint of glacial extent, we can translate the volumetric loss to a depth loss and provide an upper bound on the volumetric water loss. In reality the effect of glacier melt on the change in streamflow would be lower, which reduces the propensity to mistakenly rejecting hypothesis 6.

We also consider the potential for loss of permafrost to have produced an increase and subsequent decrease in streamflow. For example, Qiang et al. (2019) found that melting permafrost generated a temporary increase in streamflow in the upper Yellow river of 5%, corresponding to 1 cm per year of permafrost melt. To evaluate the possibility for this process in the Upper Jhelum, data were downloaded from the Global Permafront Zonation Index (GPZI) map (Gruber, 2012). Given the uncertainty in permafrost occurrence, the GPZI is presented on a scale that indicates the likelihood of permafrost, with a minimum indicating that "permafrost exists only in most favorable conditions" and maximum indicating that "permafrost exists in nearly all conditions." We binned this scale into five groups of permafrost likelihood including low, medium-low, medium, medium-high, and high. The Upper Jhelum contains no pixels with medium-high or high likelihood of permafrost and in most of the areas where permafrost is possible, the likelihood is low (see Fig. S12). To evaluate the potential for permafrost to affect streamflow, we assumed the rate of permafrost melt to be 10 mm (as in Qiang et al., 2019) from all areas with permafrost likelihood greater than zero. This provides an upper bound on the potential for permafrost melt to contribute to streamflow decline which, again, decreases the propensity of mistakenly rejecting Hypothesis 7.

### 3.4.3 Data sources: Snow

Winter precipitation occurs largely as snowfall and remains in some parts of the catchment until the late summer. Because different regions of the watershed may be affected by missing pixels (e.g., clouds) on any given acquisition date, we separated the watershed into 15 distinct zones of roughly equal areas defined by three elevation bands (<1650 m, 1650–2200 m, and >2200 m) in the five local subwatersheds corresponding to the available stream gauges. Snow contains a distinct spectral signature with high reflectance in visible and near-infrared bands and low reflectance in shortwave infrared bands, and can therefore be detected from normalized different snow index (NDSI), which is defined as (Green - SWIR)/(Green + SWIR) (Dietz et al., 2012). We generated timeseries of snow cover in each of the 15 zones using Landsat 5 imagery and Landsat 7 imagery (prior to the failure of the scan-line corrector in May 2003, see Scaramuzza and Barsi, 2005) by applying a threshold of 0.5 to NDSI to distinguish snow and water cover from dry land. We further distinguish snow (bright) from open water (dark) using a threshold of 0.2 on the NIR band reflectance (Kulkarni et al., 2002). For each zone, we selected only the dates where

missing pixels constituted less than 25% of the zone, leaving an average of 112 and 141 observations in each zone before and after 2000, respectively.

We evaluated this hypothesis by considering changes in the dates of peak snow cover and peak streamflow. The date of peak snow cover was determined using a LOESS regression (Cleveland et al., 1992) to smooth snow cover before and after 2000. More specifically, because different regions of the watershed were imaged by Landsat on different days, and in some cases parts of the watershed contained considerable cloud cover, the watershed was divided into five sub-basins (based on the stream gauges in Figure 1) and three elevation bands, creating 15 sub-regions. Within each region, LOESS regression was used to

generate a smoothed snow cover curve spanning the calendar year (combining all images for the pre- and post-2000 periods, producing two curves for each sub-region). The results were then aggregated to the entire watershed. The peak value of the smoothed LOESS curve was used to determine the date of peak snow cover. Confidence intervals for peak snow cover were bootstrapped ($N = 10^4$) by re-sampling the set of Landsat observations in each sub-region and calculating the dates of peak snow cover (pre- and post-2000) for each bootstrap iteration. We similarly bootstrapped confidence intervals on the date of

peak streamflow using LOESS regression, sampling from all of the streamflow observations before and after 2000 to estimated the "average" annual hydrograph using a LOESS regression. For each bootstrap iteration, the date of peak streamflow of the LOESS curve was extracted.

We also consider two additional analyses as robustness checks. First, we use MODIS data to estimate snow cover extent, which provides improved temporal representation of seasonal snow cover after 2000. Second, in order to account for seasonal

changes in the volumetric contribution of snow storage to streamflow, we compared total monthly snowmelt before and after 2000 using ERA5-Land data (Muñoz-Sabater et al., 2021) spanning 1984–2013. These analyses are presented in the Supplementary Information (Figs S13 and S14).

### 3.4.4   Data sources: Groundwater

We were unable to obtain *in situ* groundwater observations and remotely sensed observations from GRACE satellite were

inadequate due the large spatial averaging kernel ($\approx 40\,000\ \mathrm{km}^2$, compared to a catchment area of approximately $13\,000\ \mathrm{km}^2$) and lack of observations prior to 2002. Instead, we conceptualize the catchment as a (potentially non-linear) 'bucket' reservoir (Wittenberg, 1999; Jothityangkoon et al., 2001), in which baseflow discharge $Q_B$ is positively related with mobile groundwater storage $S$:

$$Q_B = aS^b, \tag{3}$$

where $a$ and $b$ are positive recession coefficients. Under these conditions, a reduction in baseflow can be considered indicative of a reduction in groundwater storage.

To evaluate whether baseflow declined, we apply a recursive digital filter (Nathan and McMahon, 1990) to the streamflow data to separate quickflow and baseflow (see Figure S3). We used two decades of daily flow at Baramulla station (the most downstream station) to calibrate the filter, which requires a single smoothing parameter, $\alpha$. We first set $\alpha = 0.925$ for daily

flow based on analysis from Nathan and McMahon (1990), and then tuned the value of alpha to 0.45 for three-monthly flow

so that the annual baseflow index ($Q_B/Q_{total}$) matched that of daily flow (Figure S4). Lastly, to ensure the results were not sensitive to the specific selection of $\alpha$, we re-ran the analysis with $\alpha = 0.4$ and $\alpha = 0.5$. Although the baseflow index changed, the changes were immaterial for the before-after comparisons as the results were highly correlated across combinations of all three values of $\alpha$ ($R^2 > 0.999$).

To formally evaluate Hypothesis 9, we bootstrapped confidence intervals ($N = 10^4$) to determine if there was a decline in annual baseflow in the valley in the periods before and after 2000. We also used linear regressions on quickflow, baseflow, and the baseflow index to better understand the causes of the changes in baseflow.

Additionally, to better assess our assumptions, we looked for a hysteresis in the relationship between surface water storage and streamflow. A clear hysteresis would build confidence that there is a connection between subsurface storage and streamflow, because this connection can only occur if groundwater levels are sufficiently high to reach the elevation of the lake and wetlands. A weak or non-existent hysteresis might be indicative of a weak influence of groundwater on streamflow. In contrast, a strong hysteresis would indicate both a greater influence of groundwater and the occurrence of meaningful fluctuations in the water table. Comparison between surface water extent (both Wular lake and valley inundation) and streamflow reveals a clear seasonal hysteresis (Fig. S11). We interpret these results as an indication that groundwater plays an important role mediating streamflow generation in the valley bottom.

### 3.4.5 Data sources: Surface water

A number of lakes and wetlands exist throughout the valley including Wular lake, which intersects the main stem of the Upper Jhelum between gauges (iii) and (iv), and seasonally inundated valley wetlands which capture flow from the subwatershed that drains into gauge (iv) (see Fig. 1). The actual volumetric surface water storage of the catchment is difficult to estimate. Instead, we focus on changes in surface water area using remote sensing imagery. We classify surface water extent in Wular lake and in the center of the valley in all available Landsat imagery over the period 1984–2013.

Open water is highly absorptive in short-wave infrared bands and more reflective in bands with shorter wavelengths. We use the modified normalized-difference water index (MNDWI Xu, 2006) as an indication of the likelihood of open surface water, with a threshold distinguishing between land and water pixels. Because water exhibits spatial coherence dictated by topography, we gap-filled missing pixels in Landsat 7 bands due to the scan-line corrector error by propagating edge pixels towards the center of the gap (as in Penny et al., 2018). We used a fixed MNDWI threshold across all images to classify surface water. Clouds were identified using the rudimentary Simple Cloud Score algorithm in Google Earth Engine (`ee.Algorithms.Landsat.simpleCloudScore()`) on top-of-atmosphere images. We applied this classification approach to 66 images (Landsat 5) before 2000 and 237 images (Landsat 5 & 7) after 2000 for Wular lake and valley inundation.

We formally test Hypothesis 10 by bootstrapping ($N = 10^4$) confidence intervals to compare the average water extent of Wular Lake and valley inundation before and after 2000. Seasonal analyses reveal a similar downward trend in surface water for both Wular Lake and valley inundation (Fig. S10), and images were evenly spaced across season in both the before and after time periods, meaning that our findings were not affected by a change in seasonal selection bias.

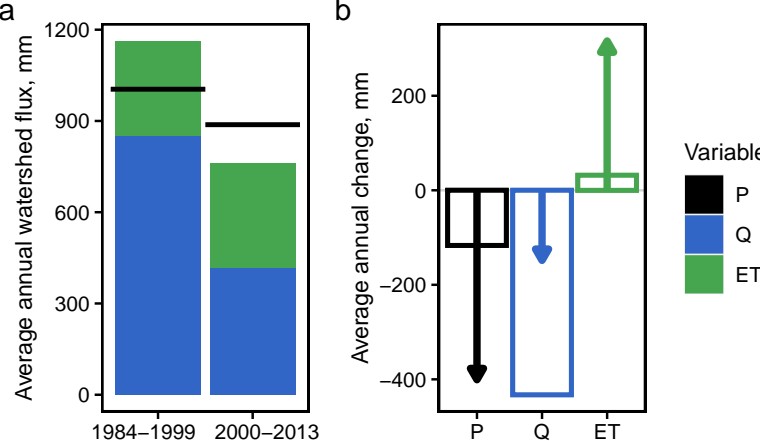

**Figure 4.** Water balance for the Upper Jhelum. (a) Average annual fluxes (P, Q, and ET) for the periods 1984–1999 and 2000-2013. (b) Average annual *change* in water balance fluxes between the two periods, given by the bars, with each flux estimated independently. The change in water balance fluxes estimated from water balance residuals is given by the arrows. In other words, each flux is calculated using the equation $\Delta P - \Delta Q - \Delta ET = 0$ and estimates of the other two fluxes.

## 4  Results

Our estimates of precipitation, streamflow, evapotranspiration, and storage change allowed the water balance to be closed for the 1984-1999 and 2000-2013 periods with an error of $\pm 15\%$ of total precipitation (Fig. 4). This suggests that the total watershed fluxes are estimated with reasonable confidence, particularly given uncertainty in rainfall interpolation and remote sensing models of evapotranspiration.

However, the analysis did not allow us to close the differential water balance (i.e., changes in the water balance) between the

two periods. We observed an average decrease in precipitation of 117 mm per year and an increase in ET of 32 mm per year, which together do not close the observed decrease in streamflow of 433 mm per year. We do not find evidence of a change in long term storage processes (e.g., decrease in glacial melt) that would close the balance. The issue, therefore, appears that biases on individual watershed fluxes (e.g., an equivalent underestimation of P and ET) might compensate each other and close the water balance at individual periods, while underestimating the components of *changes* in Q.

In order to better understand the limitations of the water balance approach, we estimated the change in each of the main water balance fluxes via water balance residual (Fig. 4b, arrows), using the values provided by independent estimates (Fig. 4b, bars) (e.g., $\Delta P_{\text{residual}} = \Delta Q_{\text{empirical}} + \Delta ET_{\text{empirical}}$). The magnitude of the change in precipitation predicted via water balance residual was considerably greater than the observed change in precipitation (-401 mm compared with -117 mm, see Fig. 4b). The residual estimate of the change in ET was much higher than the empirical estimate of the change in ET (316 mm versus 32 mm).

The magnitude of the residual for the remaining flux, streamflow, was therefore considerably less than the observed change

in streamflow (-148 mm versus -433 mm). Results were similar when using the elevation-gradient approach to precipitation interpolation (see Sect. S2, Fig. S6).

On one hand, these discrepancies highlight the uncertainty in our approach to estimate each component of the water balance individually. On the other, it illustrates the limitations of using an approach that requires water balance closure to calculate residuals or to reconcile discrepancies (e.g., predictive inference), particularly because the water balance itself does not indicate where uncertainties or biases might exist. To address this issue, we evaluate the hypotheses presented above through formal hypothesis testing (Table 2) and complementary analyses as as described above (Sect. 3). We then use the results of the analysis to construct a coherent narrative of change (Sect. 5).

**Table 2.** Summary of hypothesis tests, with 95% bootstrapped confidence intervals. Statistical significance occurs when the confidence interval excludes zero.

| Hyp. | Change in: | Conf. int | Units | Sig |
|------|-----------|-----------|-------|-----|
| 1 | Total precipitation | [-172.1, -107.2] | mm | Y |
| 2 | Winter precipitation | [ -61.5, 61.3] | mm | |
| | Spring precipitation | [-185.3, -44.8] | mm | Y |
| | Summer precipitation | [ -57.6, 30.7] | mm | |
| | Autumn precipitation | [ -37.9, 29.8] | mm | |
| 3 | Number of large events | [ -3.8, -2.5] | - | Y |
| | Number of medium events | [ -0.5, 1.6] | - | |
| | Number of small events | [ 14.1, 18.0] | - | Y |
| 4 | Temperature effect on Winter ET | [ -0.9, 1.3] | mm | |
| | Temperature effect on Spring ET | [ 1.2, 5.2] | mm | Y |
| | Temperature effect on Summer ET | [ -2.3, 1.9] | mm | |
| | Temperature effect on Autumn ET | [ -0.1, 2.3] | mm | |
| 5 | NDVI effect on Winter ET | [ -0.6, -0.3] | mm | Y |
| | NDVI effect on Spring ET | [ 7.0, 7.5] | mm | Y |
| | NDVI effect on Summer ET | [ 12.7, 13.7] | mm | Y |
| | NDVI effect on Autumn ET | [ 2.8, 3.2] | mm | Y |
| 6 | Glacier melt (upper bound) | [ NA, 0.0] | mm | |
| 7 | Permafrost melt (upper bound) | [ NA, 0.6] | mm | |
| 8 | Peak snow cover date | [ -27.0, -5.0] | days | Y |
| | Peak streamflow date | [ -28.0, 0.0] | days | Y |
| 9 | Annual baseflow | [-436.8, -241.7] | mm | Y |
| 10 | Wular lake extent | [ -24.1, -8.2] | $km^2$ | Y |
| | Valley inundation | [ -14.0, -3.0] | $km^2$ | Y |

## 4.1 Precipitation: hypotheses 1-3

Precipitation exhibited notable changes in total volume, as annual precipitation decreased by 117 mm and the 95% confidence interval excluded zero, confirming *Hypothesis 1* (Table 2). These changes were driven almost entirely by a loss of spring precipitation of 117 mm (Fig. 5). The loss of spring precipitation resulted in more uniform seasonal precipitation (Fig. 5a), confirming with *Hypothesis 2*, keeping in mind that spring was the only season with a statistically significant change in precipitation (Table 2). The other seasons saw modest and statistically insignificant changes in precipitation (on average, +20 mm in winter, -14 mm in summer, -5 mm in autumn).

There was a clear, statistically significant increase in the number of small precipitation events (2–7.4 mm) and decrease in the number of large events ($\geq 18.6$ mm), confirming *Hypothesis 3* (Table 2). Changes in medium events (7.4–18.6 mm) were statistically insignificant at the annual scale. The number of small events increased in all seasons except spring, while the number of large events decreased in spring and autumn (Fig. 5c). These results were robust to changes in the minimum event size (1–3 mm) and corresponding changes to the bins (see Sect. S2, Table S1).

## 4.2 Evapotranspiration: hypotheses 4-5

Both temperature and NDVI increased evapotranspiration in the Upper Jhelum. The changes were statistically significant for temperature in the spring and for NDVI in all seasons. This confirmed *Hypothesis 4* and *Hypothesis 5* (Table 2), noting that the only statistically significant increase in temperature occurred in spring (+1.9°C). There were statistically insignificant increases in temperature in the remaining seasons including winter (+0.8°C), summer (+0.4°C) and autumn (+1.0°C, Fig. 5).

Annual average watershed evapotranspiration increased by 32 mm, from 311 mm before 2000 to 343 mm after 2000. Although this change is smaller than the uncertainty in water balance closure (i.e., 15% of precipitation), the results are statistically significant and the direction is in agreement with the estimate of ET from water balance residual.

The two hypotheses pertaining to ET seek to attribute this increase to either increasing temperature or changing land use. As noted in Fig. 3, however, it is possible that land use may also affect temperature (e.g., through radiative forcing and the sensible and latex heat fluxes) and that temperature can also affect NDVI (e.g., through changing phenology) (Figure 3). To better evaluate these factors, we further consider changes in ET within different land use classes (Fig. 6).

The most dramatic increases in ET occurred within agricultural land cover classes (cropland and mosaic vegetation, i.e. orchards), which constituted 27% of the catchment area in 2001. In these classes, NDVI increased substantially in the valley in the spring and summer seasons (Fig. 6ab), corresponding with the primary growing season for paddy, maize, and orchards. Between 2001 and 2011, the catchment exhibited notable expansion of the mosaic land cover class, including approximately 230 km$^2$ converted from traditional crops to mosaic. The largest local increases in ET are associated with the expansion of the mosaic class (see Fig. 7a), with ET increasing by 70 mm (mosaic to mosaic), 78 mm (cropland to mosaic), and 82 mm (shrubs and grassland to mosaic).

Noticeable increases in ET also occurred in the large portion of the watershed area (53%) that was consistently classified as shrubs and grassland in both 2001 and 2011. NDVI increased along the hills on the southwest and northeast portions of the

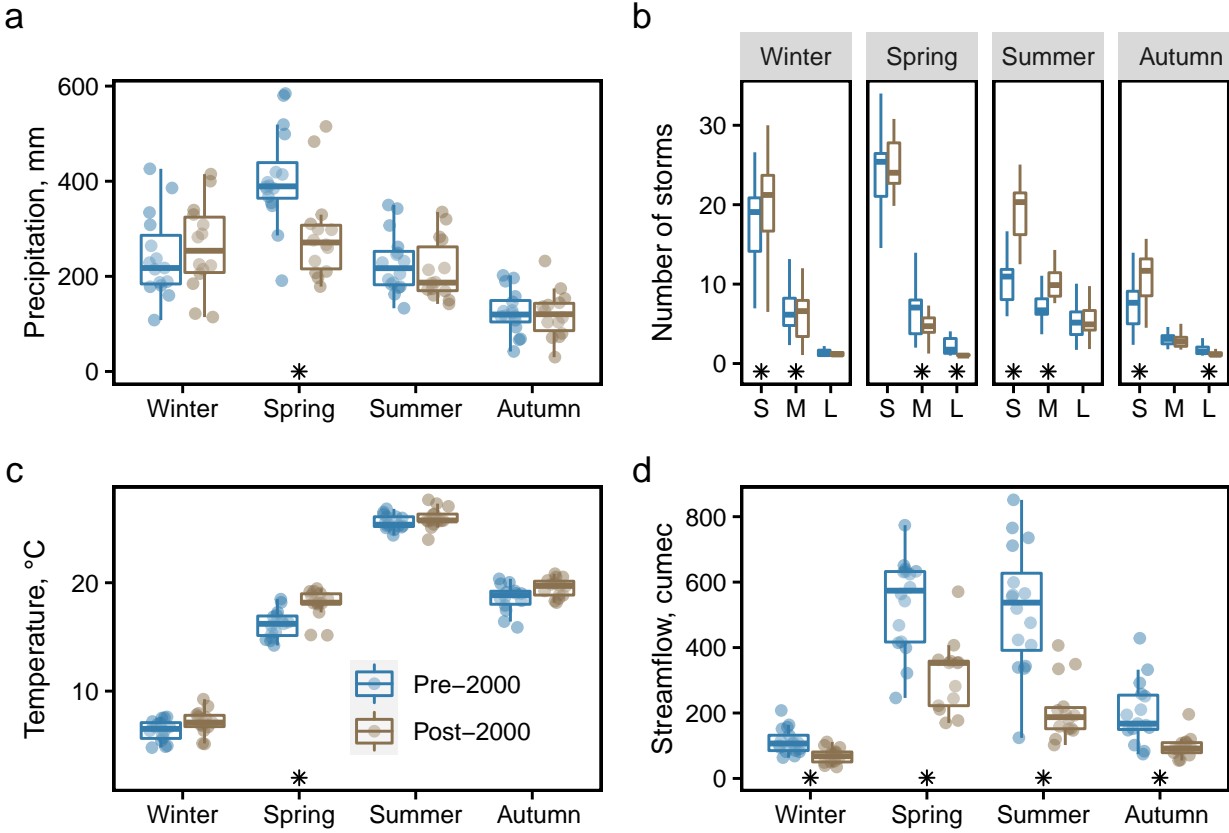

**Figure 5.** Seasonal climate and hydrology in the Pre-2000 (1984–1999) and Post-2000 (2000-2013) periods including (a) mean seasonal gauge precipitation, and (b) mean number of storms in PERSIANN grid cells for small (S: 2–7.4 mm), medium (M: 7.4–18.6 mm), and large (L: $\geq$18.6 mm) events, (c) mean gauge temperature, and (d) mean streamflow in cumec ($m^3\ s^{-1}$) at the watershed outlet (gauge v, Baramulla station). Statistically significant differences are noted by ($*$) when the 95% bootstrapped confidence intervals exclude zero. Notably, spring exhibited dramatic changes in temperature (+1.9°C) and precipitation (-117 mm). The number of small storms increased across all seasons, while large storms decreased in all seasons except summer.

watershed, resulting in higher ET in grassland / shrubs and forest land cover. NDVI remained constant in the Wular lake in spring but increased considerably in summer, likely due to increasing fertilizer application supporting algae and other aquatic vegetation in the lake (Wetlands International South Asia, 2007). In contrast, regions where NDVI appears to have decreased are dominated by urbanization in the center of the valley (visible in Fig 6ab) and mountain peaks with near-constant cloud cover in the summer, which occur along the southeastern and northwestern watershed boundaries. In these pixels, few ($\leq$5) summer NDVI observations are available before 2000 due to cloud cover, and these piexels exhibit a downward bias in the

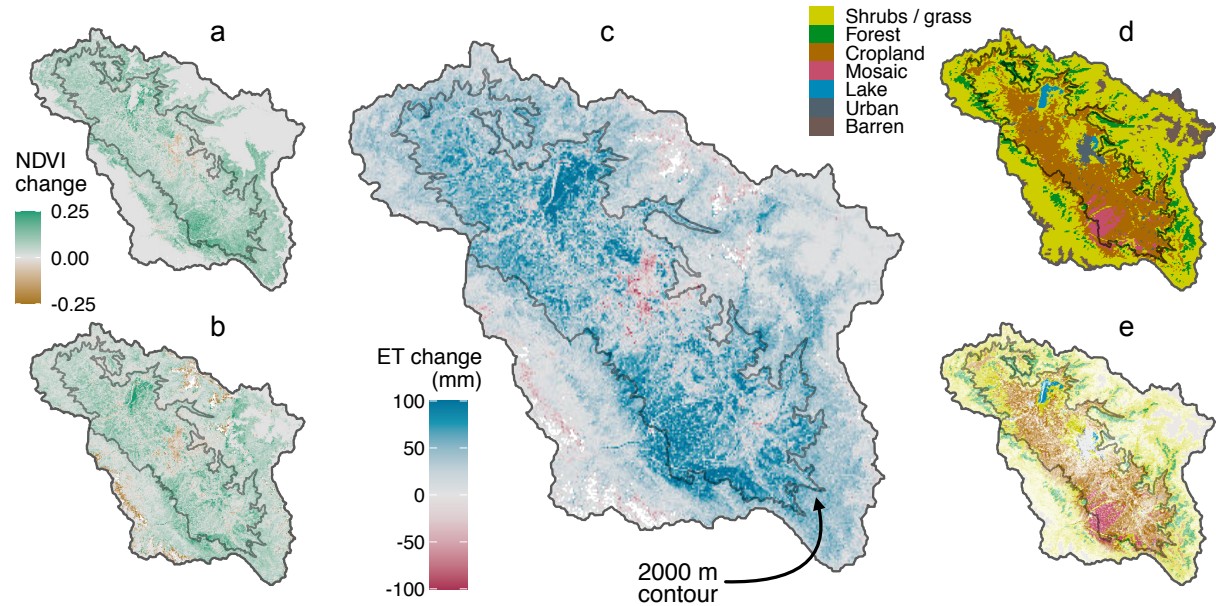

**Figure 6.** Spatial changes in evapotranspiration after 2000. (a) Spring and (b) summer changes in NDVI. (c) Annual change in evapotranspiration (ET). (d) MODIS land cover map from 2010. (e) The same MODIS 2010 land cover map, with pixel transparency determined by the magnitude of positive change in annual ET from (c). The 2000 m contour separates the low-elevation valley from moderate elevation hillslopes in both mountain ranges. Changes in ET are clustered in the valley, especially in cropland and mosaic vegetation (i.e., orchards), with increasing ET in natural vegetation just above 2000 m. See text for details.

reported NDVI change relative to pixels with more images (see Fig. S8). This suggests that any potential bias would lead to a conservative estimate on the change in ET.

On average, of the 32 mm annual increase in ET that we detected, approximately 17% can be attributed directly to increasing air temperature through its effect on PET, and the remaining 83% to an increase in NDVI. The largest net contributors to watershed-averaged increases in ET were shrubs and grassland (15 mm), cropland (11 mm) and forest (3 mm). Although associated with strong local increases in evapotranspiration, mosaic vegetation covered only 2.7% of the watershed in 2011 and only contributed a 2 mm increase to watershed-average ET. The black boxes in Fig 7b encompass the change in ET for

regions of the watershed that maintained consistent land cover in 2001 and 2010, accounting for a total increase in ET of 27 mm compared with 5 mm in regions where land cover changed. We can therefore infer that warming temperatures not only had a clear effect on watershed evapotranspiration through the direct effect on PET, but also indirectly by increasing NDVI within naturally vegetated land classes. This indicates that our findings for *Hypothesis 4* are likely conservative. Regarding *Hypothesis 5*, land use change has led to large local increases in ET, but the watershed-average effect of land use indicated by

Table 2 is likely to be overestimated. This is because the overall effect on the catchment water balance in pixels where land use changed is small compared to the effect in pixels where land use remained the same. Nevertheless, it is possible there were

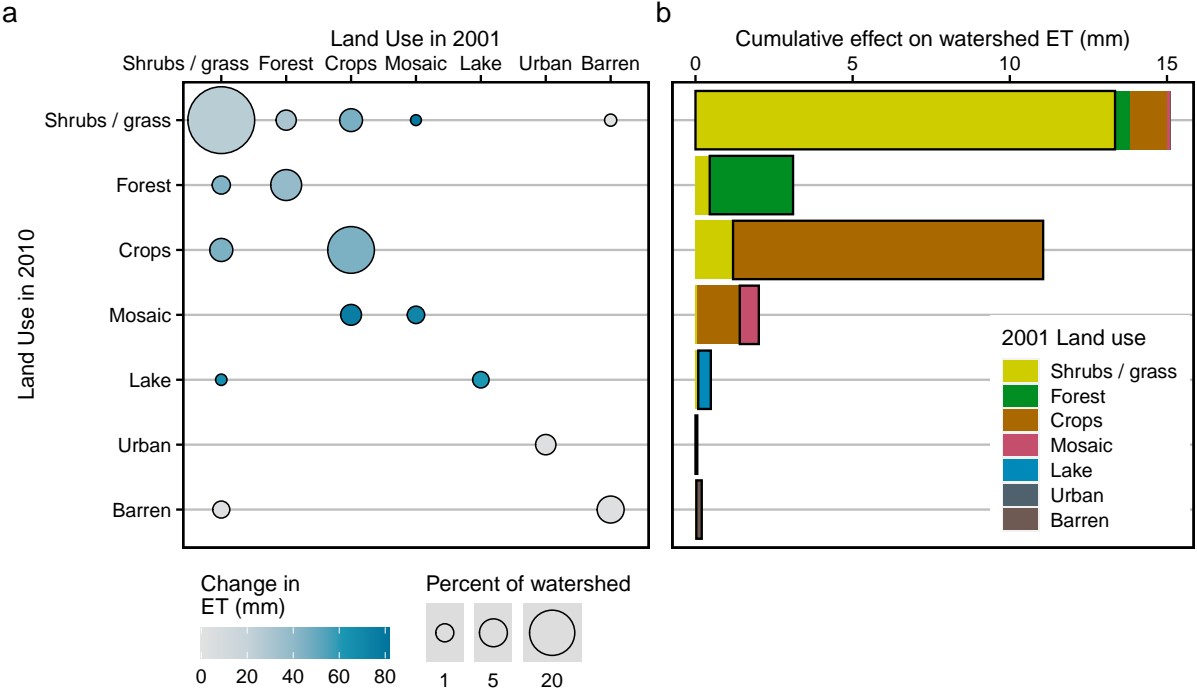

**Figure 7.** Changes in land use and evapotranspiration. (a) Change in evapotranspiration within each combination of land use categories from 2001 and 2010. The size of the circle represents the fractional area covered by each pairwise grouping, and the shading of the circle indicates the average change in ET per unit area. For instance, the largest increase in ET occurred in pixels that changed from crops in 2001 to mosaic in 2010 (+80 mm), but this represented a small fraction of the watershed (<5%). ET increased in every category. (b) The cumulative change in ET by 2010 land use category (note that the y-axis continues across both panels). The cumulative effect (or watershed-average) ET is equal to the *Change in ET × Fraction of watershed* for each land use combination. For instance, the pixels that changed from Crops in 2001 to Mosaic in 2010 reduced ET by 1.4 mm when averaged over the entire watershed. Overall, most of the increase in watershed ET (27 mm) occurred in regions where land cover remained consistent from 2001 to 2010 (b, see bars outlined in black), compared with regions where land cover changed (5 mm, no outline).

additional changes in land management (e.g., increased irrigation) that could have affected these results and were not captured by the land use classification from Strahler et al. (1999).

### 4.3 Catchment storage: hypotheses 6-10

Although a permanent loss of frozen water storage in the Upper Jhelum might play an important role in the water balance in some tributaries, the overall effect on the watershed as a whole was small. The upper bound on average annual glacial contribution to streamflow was 0.003 mm, and the upper bound for the permafrost contribution was 0.64 mm. We reject *Hypothesis 6* and *Hypothesis 7* on the basis that the effect on watershed streamflow is likely negligible. The primary reason

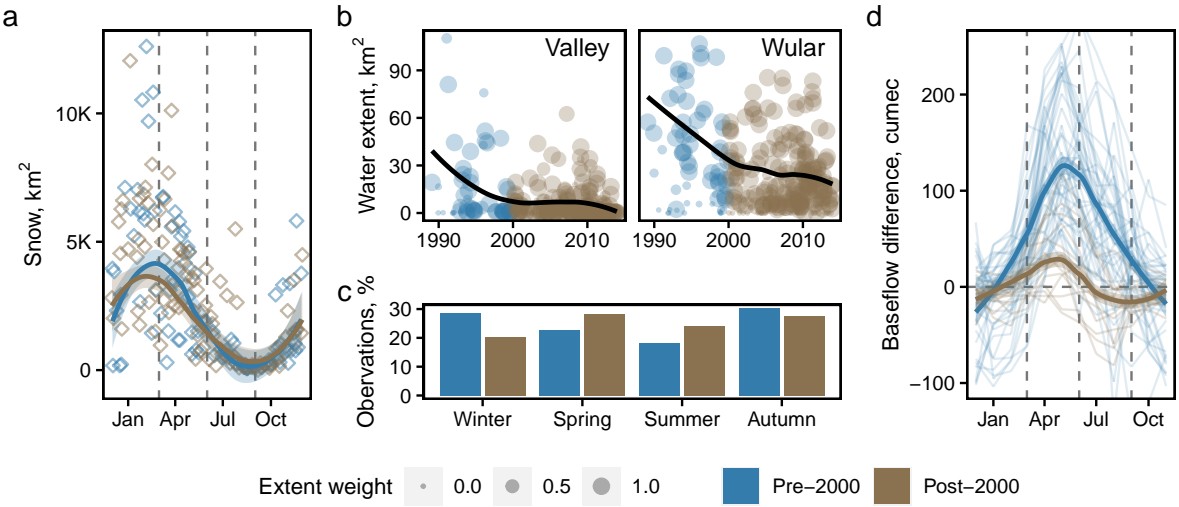

**Figure 8.** Declining water storage and baseflow. (a) Landsat observations of snow cover with loess smoothing, highlighting earlier spring snowmelt after 2000. (b) Long-term trends in valley inundation and Wular lake indicating decreasing surface water storage over time. The loess smoothing of water extent is weighted by cloud cover given by $\exp(-A_{cloud}/A_{total})$. (c) The percent of satellite images used to assess water extent was seasonally consistent before and after 2000. (d) Baseflow at gauge iv. (Sopore) minus baseflow at gauge iii. (Asham), encompassing the river reach that includes Wular lake. After 2000, baseflow peaks and depletes earlier in the year, as does a transition from gaining to losing conditions.

these contributions to streamflow are so small is due to the limited spatial extent of glaciers (1%) and permafrost ($\leq 6.4\%$,
Fig. S12) relative to the entire area of the Upper Jhelum watershed.

We found statistically significant changes in the seasonal timing of peak snow cover and peak streamflow. After 2000, peak snow cover occurred 15.4 days earlier and the confidence interval included zero. Peak streamflow occurred 13.2 days earlier on average, and the upper limit confidence interval was equal to zero. This provides confidence that the earlier peak in snowmelt led to an earlier peak in streamflow, confirming *Hypothesis 8*. We note that observational data was missing to quantify the
effect of changes in snow storage volumes. However, reanalysis data from ERA5-Land indicates that snowmelt increased in earlier months and decreased in later months after 2000, at all elevations in the watershed (see Fig. S13). Data from MODIS also shows an earlier peak in snow cover after 2000 (Fig. S14).

We found statistically significant changes in baseflow exiting the watershed (Table 2), which is consistent with the reduction in groundwater storage in the valley described in *Hypothesis 9*. We complement this result by presenting additional findings
pertaining to baseflow and groundwater–surface water interactions in the watershed. First, we compare temporal trends in streamflow below Wular lake (gauge iv., Sopore station) and the most upstream gauge of the watershed (gauge i., Sangam station). The streamflow timeseries at both locations exhibited statistically significant decreasing trends over the period 1960–2013 (Fig. 9a). Baseflow separation, however, revealed important differences between both streamflow timeseries. Baseflow

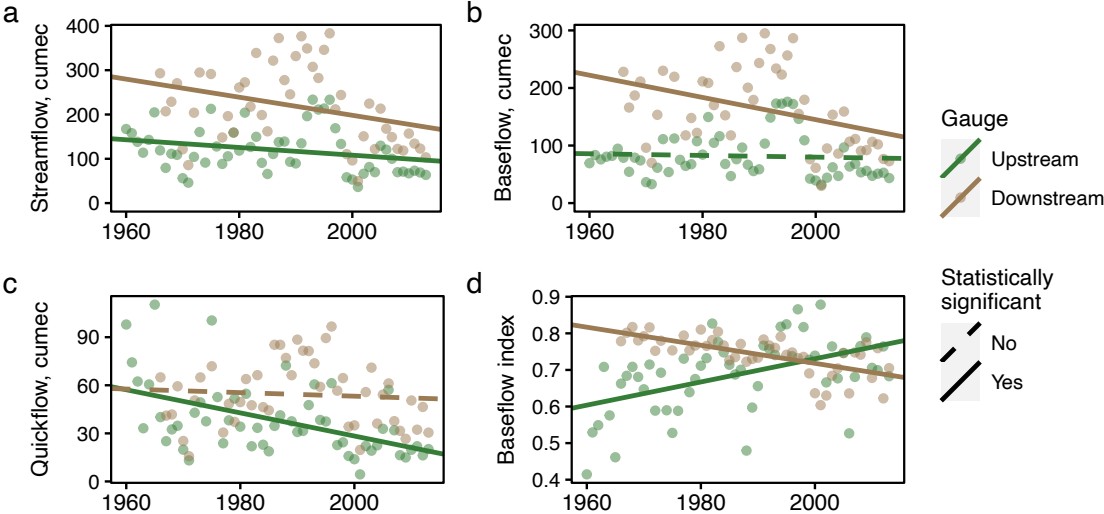

**Figure 9.** Long-term trends at Sangam station (upstream, gauge i.) and Sopore station (downstream, gauge iv.) of (a) streamflow, (b) baseflow, (c) quickflow, and (c) the baseflow index, which is the fraction of streamflow comprised of baseflow. The statistical significance in this analysis was associated with $p < 0.1$ for a nonzero trend from a linear, least-squares regression. All statistically significant trends were also significant for p < 0.05 except for the Upstream trend in panel (a).

decreased over time only in the downstream gauge (Fig. 9b) whereas quickflow decreased only in the upstream gauge (Fig. 9c).

Temporal changes in the baseflow index ($B = Q_B/Q_{Total}$) of each of these gauges therefore occurred in opposite directions, with decreasing $B$ near the outlet and increasing $B$ in the hinterlands (Fig. 9d). This lends additional credence to *Hypothesis 9*, and we further discuss potential causes and implications of these opposing trends in the baseflow index with respect to saturated and unsaturated groundwater storage in the Discussion (Sect. 5).

Classification of surface water reveals that surface water storage declined dramatically during the study period, both in Wular

lake and the neighboring wetlands (Fig. 8b), confirming *Hypothesis 10*. Both of these surface reservoirs connect to the main stem of the Upper Jhelum between gauges iii. and iv. (see Fig. 1), and likely play an important role in streamflow generation along this reach.

There appear to be clear relationships among surface water storage, groundwater storage, and streamflow. A comparison between surface water extent and streamflow exhibited a clear seasonal hysteresis (Fig. S11). We also computed locally generated

baseflow as the difference in baseflow between gauges iii. and iv, a reach which contains Wular lake and the valley wetlands. Baseflow along this reach peaks earlier and at a much lower amplitude after 2000 (Fig. 8c). It also transitions much earlier (mid-summer) to losing conditions (i.e. negative local streamflow values), compared to pre-2000 when baseflow transitioned to losing conditions only at the end of autumn and beginning of winter. Taken together, these findings suggesting that storage plays a critical role in mediating streamflow in the Upper Jhelum.

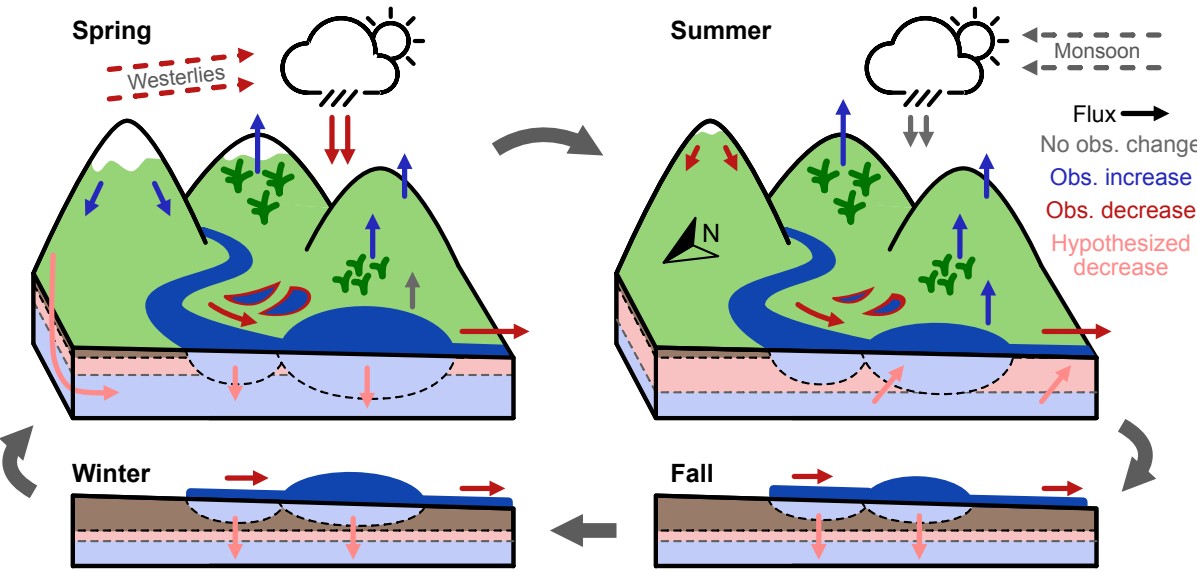

**Figure 10.** Conceptual diagram of seasonal water fluxes and direction of change. The subsurface cross section represents the thick layer (up to 1 km) of sedimentary deposits in the valley (see Sect. 2). Black, red, and blue arrows indicate observed fluxes combined with the observed sign of change. Pink arrows indicate groundwater fluxes that have been inferred and hypothesized to decrease over the study period. Both natural vegetation on hillslopes and agricultural crops in the valley exhibit increased evapotranspiration (Fig 6). See text for details.

## 5   Discussion

We now synthesize the results presented above to develop a narrative of hydrological change by reconciling the various fluxes that have either increased or decreased over time (Fig. 10, red and blue arrows). Taken together, the observed changes in hydrological fluxes indicate additional unobserved changes in fluxes connecting surface and groundwater that might play an important role in explaining hydrological change in the catchment. As discussed in the following paragraphs, evidence suggests that these fluxes have decreased over time (Fig. 10, pink arrows).

- The season with the most precipitation is spring, when the prevailing climate is driven by westerlies, yet spring precipitation declined considerably during the study period. At the same time, vegetation activity increased across most of the watershed within both anthropogenic (cropland, orchards) and natural (forest, shrubs and grassland) land use classes (Fig 6a), producing an increase in evapotranspiration. The corresponding reduction in spring streamflow may have been partially compensated by higher temperatures and earlier snowmelt. Before and after 2000, snowpack storage was mostly exhausted by the end of spring (Fig 8a).

- Springtime seepage generates high groundwater recharge throughout the valley, both from the highlands and from within the valley (Jeelani, 2008). High groundwater levels at the end of the spring season are reflected by the peak in baseflow

along the reach of the stream that includes Wular lake (Fig. 8d). Prior to 2000, this peak occurred near the transition
between spring and summer. After 2000, this peak occurred earlier in spring and at a much lower level of baseflow.
This suggests considerably lower groundwater storage over time, reflected in the reduced groundwater recharge fluxes
(Fig 10, Spring).

- In summer, the prevailing climate is driven by the Indian monsoon and precipitation is generally less than westerly
  precipitation in the spring (Fig 5a). Consequently, hydrology within the watershed is controlled largely by water storage
  (snow, lakes and groundwater) left over from spring. Prior to 2000, the seasonal recession of the baseflow hydrograph
  starts high at the end of spring and continues to produce discharge throughout the summer season (Fig 8d). The river
  then transitions to losing conditions in autumn and early winter before baseflow increases with winter precipitation and
  snowmelt (Fig 8d). After 2000, surface water (Fig. S10) and groundwater (Fig. 8d) storage are greatly reduced at the
  beginning of summer. The receding limb of the hydrograph starts low at the beginning of summer and quickly depletes,
  transitioning to losing conditions within the summer season. Indeed, the recovery of local baseflow in autumn (i.e.,
  baseflow becoming less negative, see Fig. 8d, brown) is driven by declining baseflow downstream rather than increasing
  baseflow upstream. Additionally, summer evapotranspiration after 2000 increased throughout much of the watershed
  including in natural and anthropogenic land use classes.

- The hydrological cycle in the watershed is mostly dormant in autumn and winter. In autumn, little precipitation falls and
  the hydrograph recedes into winter. Notably, lake storage depletes and the lake transitions to losing conditions in several
  years, particularly before 2000 (Fig. 8d). Winter precipitation arrives mostly as snowfall, replenishing snow storage.
  Winter rain and snowmelt serve as the early primers for the seasonal cycle to renew in spring.

To summarize, climate appears to be the primary cause of hydrological change within the Upper Jhelum. The most influential
driver is the decline of spring westerly precipitation. Other studies have associated this decrease with warming temperatures
and climate change (Zaz et al., 2019). This effect is compounded by an array of other drivers that affect watershed processes.
Notably, the loss of baseflow downstream of Wular lake suggests a decrease in groundwater storage in the valley. This decline
in groundwater is facilitated both by reduced rainfall during the spring and increasing watershed evapotranspiration. The latter
might be supported by an *increase* in the number of small precipitation events that are less likely to generate runoff. This
observation from the PERSIANN CDR dataset allows us to hypothesize that the shift towards a larger number of smaller
storms results in reduced overland and macropore flow, along with more stable and persistent soil moisture and ultimately
more water "lost" to vegetation uptake. In this case, seepage would be increasingly likely to occur via slow drainage processes,
rather than macropore flow activated in large storms. Such changes have been observed in other karst catchments (Zhao et al.,
2019) and are supported by the evidence that quickflow declined and the baseflow index increased in the most upstream gauge
in the Upper Jhelum (Fig. 9).

The increase in evapotranspiration appears to have occurred throughout the catchment. Our evapotranspiration model indi-
cates that increasing air temperature had a small direct effect on ET via PET (17% of the total increase) and that most of the
overall increase in ET (83%) occurred due to changes in NDVI, which increased in all natural and anthropogenic vegetation

classes. Evapotranspiration exhibited the greatest increases in regions that transitioned to orchard cultivation, but these areas represent a small fraction of the overall watershed and increase in ET. In places where land use was unchanged, the observed increase in NDVI indicates an increase in water availability (when ET is water limited), an increase in energy availability (when ET is energy limited), or an increase in nutrient availability (where plant growth is limited by nutrient availability). In other words, such greening could arise due to agricultural intensification (via increased irrigation or fertilizer application) or increasing temperature due to climate change. These findings are consistent with other studies. For instance, agricultural intensification has been documented in the Upper Jhelum (Wetlands International South Asia, 2007) and modeling studies have demonstrated that warming temperatures will produce more favorable conditions for plant biomass growth (Rashid et al., 2015).

## 6  Conclusions

In this study, we develop an empirical approach to hydrological attribution using the method of multiple hypotheses to understand the drivers of dramatic changes in the annual streamflow of the Upper Jhelum river. We find that much of the observed decrease in streamflow is associated with decreases in westerly precipitation in spring, in addition to greater evapotranspiration. While land-use change to orchard plantations and agricultural intensification are likely contributing factors, we attribute most of the increase in evapotranspiration to non-local anthropogenic causes, most notably increased vegetation activity in spring, likely due to increased temperature and earlier snowmelt. Changes in key fluxes of the water balance (P, ET, and $\Delta$S) do not fully account for changes in streamflow (Q), and there remain considerable differences between the change in fluxes from independent estimates and the change estimated from water balance residuals.

By focusing on the cumulative understanding from evaluating separate but complementary hypotheses, we nevertheless develop a coherent narrative of hydrological change in the Upper Jhelum basin. In addition to the loss of westerly precipitation, there appears to be a reduction in groundwater storage evidenced by the considerable reduction in baseflow in the valley. This situation contrasts with upstream changes, where declining streamflow occurred primarily through reductions in quickflow, and could potentially be explained by changing precipitation patterns. These findings suggest multiple directions to guide future research in the basin, including better characterization of (a) baseflow generation and surface water–groundwater interactions, (b) the role of soil moisture, phenology, and rainfall intensity in mediating the water balance on hillslopes outside the valley, and (c) vegetation water consumption in both natural and human land uses within the watershed.

The method of multiple hypotheses is a promising tool to attribute hydrological change in situations where process uncertainty might be compounded by hydrologic regime shifts. While outcomes associated with individual hypotheses might exhibit considerable uncertainty, especially in data-scarce catchments, together the multiple hypotheses provide multiple strands of evidence to support (or refute) specific mechanisms and ultimately attribute hydrological change.

*Data availability.* The secondary data that support the findings of this study are available on request from the corresponding monitoring and collection agencies. Remote sensing products used in this study are freely available from the respective providers (see in-text citations).

*Author contributions.* G.P., Z.A.D., and M.F.M designed the research, G.P. conducted the analyses, G.P. and M.F.M wrote the paper

*Competing interests.* The authors declare that they have no conflict of interest

*Acknowledgements.* G.P. and M.F.M. acknowledge support from the National Science Foundation under Grant No. ICER 1824951.

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
