# Peer review of "Climatic and anthropogenic drivers of a drying Himalayan river"

_Hydrology and Earth System Sciences, 2021_

## Author Comment (AC1)

Response to Reviewer 1

Reviewer comment: https://doi.org/10.5194/hess-2021-274-RC1

Reviewer comments are in black. Author responses are in blue.

I read this paper with interest. It deals with an issue that is becoming more and more important, i.e., hydrologic change. Increasingly we are having to attribute possible causes of change and to develop models that can reproduce past change and predict change in the future. In this paper the authors are dealing with the Upper Jhelum catchment in the headwaters of the Indus River, which has experienced decreasing streamflows. The paper is aimed at attributing possible causes of the decreasing streamflow. They implement the notion of multiple working hypotheses to assess a combination of alternative causal explanations for this observed decrease. I found the analyses to be insightful and valuable, albeit somewhat inconclusive due to data limitations and uncertainty.

While the paper is publishable (eventually), I found the presentation to be confusing and unnecessarily diffuse.

We would like to thank the reviewer for considering the manuscript and providing constructive and helpful feedback. As the following three comments all relate to the framing of the paper, we address them together, below.

Firstly, for some reason, the authors have conceived the paper as a methodological advance, and have gone to a lot of trouble to give an overly philosophical introduction, talking about top-down and bottom-up and multiple working hypotheses etc. I had to read four manuscript pages before they introduced the problem they are studying. I would like them to start with data evidence of both the cause and effect, before presenting the alternative hypotheses as part of their methodological presentation.

Secondly, this is a place-based study, the sooner you get to the problem, the better its readability. The philosophical discussion can be kept to a minimum. I would like to know more about the streamflow decrease as early as possible, and the history of climatic, and anthropogenic changes to land use and land cover as possible causal factors. In other words, I like more of an introduction to the basin and its hydrology that can help motivate why it is that they form their multiple hypotheses.

Thirdly, I am not even sure if the authors are correct in calling their modeling approach a bottom-up study. The way I read it, the authors analyze data at the catchment scale and find a decreasing pattern of streamflow. They then systematically try to attribute it to change in rainfall, evapotranspiration, land use and land cover change etc, or a combination of these. In my opinion, this is in fact a data-based, top-down study. I go back to the original source (Klemes, 1983), who defines the downward (top-down) approach "starts with trying to find a distinct conceptual node directly at the level of interest (or higher) and looks for the steps

that could have led to it from a lower level".  This is the way I see the authors' attribution work.

Thank you for these comments. We agree that the paper focuses primarily on understanding the hydrology of the Upper Jhelum. We will therefore increase the emphasis of the Upper Jhelum within the introduction, and reduce the emphasis on the methodological approach. With that said, we do not want to relegate the approach entirely to the methods section, because we believe it presents an alternative to the many emerging studies of hydrological attribution that rely on predictive inference (e.g., basin hydrological simulation), particularly when focusing on watersheds at spatial scales as large as the Upper Jhelum. We had hoped to emphasize this contrast by referring to our study as a "bottom-up" attribution. However, we now recognize that this terminology is confusing and should be avoided.

In response to these comments, we will reduce the scope and length of introductory paragraphs, so that we briefly describe the importance of hydrological attribution and review existing literature that focuses on attribution. This will allow us to better situate the changing hydrology of the upper Jhelum in response to ongoing human activities including both drivers and outcomes. In providing greater emphasis on the Upper Jhelum, we will also discuss the decline in streamflow in greater detail along with an expanded literature review on potential causes and mechanisms for the drying of the upper Jhelum.  We hope this will provide better framing for our approach, hypotheses, and subsequent analysis.

Finally I would like to see a clear statement of their conclusions, focusing more on the hydrology of the place and perhaps less on methodological sophistication. In my opinion, the paper raises a number of important issues about the hydrology of this region, and placing the result in the context of the local hydrology would be much more useful than whether this is a bottom-up or top-down method.

Per your suggestion, we will de-emphasize the methodological components throughout the paper (including removing references to bottom-up and top-down) and focus the paper more on the implications for local hydrology. Additionally, in the conclusions, we will better situate the methodological discussion as part of a broader discussion on the hydrology of the Upper Jhelum, including key questions for future research.

The paper is definitely publishable after major revisions to address or discuss (in case the authors want to dispute my comments) the issues I have raised in my review

---

## Author Comment (AC2)

Response to Reviewer 2

Reviewer comment: https://doi.org/10.5194/hess-2021-274-RC2

Reviewer comments are in black. Author responses are in blue.

The Penny et al. manuscript presents a variety of observations to investigate why the water balance for a Himalayan river has changed. They highlight the limits to a simple water balance calculation and instead include information on land-use, NDVI and water extent as markers for differences in vegetation or crop productivity (i.e., evapotranspiration) and groundwater contributions to surface water, respectively. They focus on the changes between a 15-year (1984-1999) and a 13-year (2000-2013) period, and approach the work as a 'method of multiple hypotheses', testing various hypotheses related to potential drivers of the change in streamflow. I found the manuscript relatively well-structured and I think that it can be a good addition to the literature and a good resource for water managers in the Jhelum river region. However, I do think the manuscript needs some work before it is ready for final publication.

Thank you for the careful consideration and constructive feedback, which we believe will strongly improve the manuscript. We respond to each of your comments below.

From a clarity perspective, I found that some words were not well defined ("secondary data", "drivers", "changing climate") and that the reader would benefit from describing these better. From a structural perspective, I found that the methods and results were not always matching up. How was the baseflow index calculated? How is a storm event characterized? Why is a basin-average correct value for precipitation preferred over a locally correct value? How were statistical analyses performed?

In the revised manuscript, we will clarify each of these terms upon their first use. The approach for calculating for baseflow was previously relegated to the SI, and we will port this to the manuscript. The remaining questions are raised in the minor and specific comments below, and we address them individually under each comment.

What was also clearly missing in the manuscript were uncertainty estimations on the data used to test the various hypotheses. The authors indicate an uncertainty in their initial water balance estimation of 15% (which by the way is more than increase in ET and almost as much as the decrease in streamflow of 117 mm?). An estimation of the uncertainty related to individual components of the water balance would be helpful here – or is it 15% of each component? Uncertainty estimations were given for none of the following analyses… (precipitation estimations from remotely sensed observations, evapotranspiration estimates using crop factors, NDVI estimations when only few (=< 5) images were available, and inferring catchment storage from surface water extent). I expect there to be considerable

uncertainty in each of these estimations, and expect that to be quantified and discussed in the text of the updated manuscript.

Thank you for this comment. Estimating uncertainty represents an important challenge in this study. Before describing how we address this comment, we wish to emphasize that the manuscript is oriented towards improving conceptual understanding of hydrological change. A key objective of the paper is to demonstrate the use of multiple hypotheses as an alternative approach for attribution in situations where data scarcity and uncertainty confounds the application of a model-based predictive inference approach. One important reason we included the water balance was to give a sense of the overall uncertainty with respect to water balance fluxes and therefore inform interpretation of the overall approach. That being said, we completely agree that observation uncertainties should not only be used to justify our approach, but also systematically incorporated into the approach itself.

With these clarifications in mind, we will improve our analysis of uncertainty in three ways, with an emphasis on providing a stronger statistical foundation for the evaluation of the hypotheses.

1. "Testing" each alternative hypothesis implies determining whether the signal is stronger than the noise. We will run statistical tests for hypotheses 1-5, and 8-9. In cases where we have confidence that errors are normally distributed, we will use a t-test and will apply a Wilcoxon signed-rank test otherwise. This will give greater confidence in our evaluation of each individual hypothesis. Hypotheses 6 and 7 deal with glacier and permafrost contributions to the change in streamflow. For these hypotheses we only have upper and lower bounds, and therefore instead of traditional hypothesis testing we will provide upper and lower estimates on the expected changes.
2. We will better describe uncertainty due to spatial variability of precipitation by showing a map of the differences in the two interpolation methods. We will also better describe how using the different regionalization approaches would affect our hypotheses. In particular, although the elevation-gradient interpolation yields a much larger estimate of annual precipitation, the results of the before-after comparison are consistent regardless of which dataset is used.
3. As an independent estimate of the uncertainty associated with each component of the water balance, we will compare our findings to what would be expected from a water balance closure. For example, we will compare our estimate of ET based on NDVI and temperature to its water balance estimate (P-Q-$\Delta$S)  and describe how that might change our conclusions.

Ultimately, the benefit of the multiple hypotheses approach is that the findings from some hypotheses can corroborate or cast doubt on the findings from other hypotheses. The fact that the findings from many of the hypotheses lead to conclusions that are consistent gives credence to the findings that we present.  With that said, we agree that a more careful

treatment of uncertainty will not only provide greater confidence in the approach but greater context for readers interested in the Upper Jhelum, and we are thankful to the reviewer for this suggestion.

Detailed comments:

Title: What do the authors mean with 'secondary data'? – being specific would be helpful here to guide the potential reader to reading the article.

We will clarify that 'secondary data' refers to datasets collected by other sources -- in this case, mostly governmental agencies. The reason the emphasis on secondary data is important is because collecting primary hydrological data in remote (or transboundary) locations is often impractical, and therefore reliable approaches that build on publicly available data sources can support effective water management.

L28 Instead of pointing out that there are many watersheds where the hydrological drivers are unknown, which is not surprising given the amount of watersheds globally, highlighting prior studies that did identify and quantify drivers would be more helpful.

Thank you -- in response to this comment and comments by the other reviewer, we will condense this part of the introduction to focus on existing approaches and studies for attribution, and note how these studies might inform our attribution study.

L34 "Their associated drivers" which drivers are meant here?

We will clarify that by "associated drivers" we mean, broadly, the "causal drivers of hydrological change".

L76 "finding"s please add the s

L78 remove "by"

Thanks for catching these grammatical errors and typos.

L124-129 consider removing this text since a standard manuscript format is followed, or adapt such that it contains information that is specific to this manuscript.

We will address these comments in the revised manuscript.

Fig 1 The panels 1d-g are square, but they are not square where they are indicated on map 1b. Did the coordinate system change? Then please indicate the new coordinates along insets d-g. If not, please make sure that the indication and maps match. Also, for me it

would have been helpful to have the line of the river drawn in panels d-g, and the river and catchment boundary are not shown in the legend (I assume light blue and black lines).

We will fix the projection in this figure so that the shapes of the subpanels are consistent and we will also include the main river channels and watershed boundary.

Fig 2 snow density is needed to calculate water content from area and depth, and should be included in this scheme as well as in the calculation.

We will clarify in the text that we do not calculate snow storage volumes, but rather focus on snow areal coverage. This is consistent with hypothesis 8, which states that "Reduced snow cover and earlier snowmelt generated an earlier peak in annual hydrograph." Although estimating snow storage volumes would be beneficial, we believe the benefits would be overshadowed by the various assumptions needed to estimate snow depth (ie, the need to spatially and temporally interpolate daily snowfall depths and daily temperatures, and model snowmelt and sublimation). As noted in response to another comment, we will present the dates of peak snow cover and streamflow.

L173-174 define "changing climate" and justify how a dataset that is shorter than 30 years can be representative of a change in climate.

We clarify that we mean by "changing climate" a shift in weather patterns -- that may or may not be associated with climate change.

L178 'a reduction in storm size' is not captured in the hypotheses yet – greater storm frequency does not always mean in a reduction in storm size. Also, there was no explanation on how 'storms' were characterized (precipitation magnitudes separated by an interval of how many hours at least?). Lastly, in many parts of the world a 'storm' can just be a period with a lot of wind, and does not necessarily imply rainfall. Please consider changing it to 'rainfall event' or 'precipitation event'.

Thanks for this comment. The reason we originally focused on storm frequency is due to the fact that remote sensing products are quite reliable for observing the occurrence of precipitation, but have high uncertainty when estimating the depths of storms (especially in mountainous regions). As stated in the manuscript, we use remotely sensed observations of daily precipitation because our gauge date is limited to monthly observations. To address this comment, therefore, we will convert frequency precipitation frequency to average event depth by combining the frequency observed from PERSIANN with the monthly total precipitation from gauges. We will also change the terminology of "storm event" to "precipitation event" and clarify that a precipitation event is any day in which precipitation was ≥ 1 mm.

L181 The authors hypothesize here about greater bare soil evaporation, but included no information on soil moisture (only groundwater), neither did they explain why this information is not considered.

We are unaware of any datasource that would provide reliable information about soil moisture before and after 2000. For this reason we can only hypothesize what could have changed. The intention of mentioning soil moisture in this passage is to describe how a change in storm frequency could affect the water balance (as has been observed in other watersheds). We will therefore clarify in the text that this is only a possible explanation that would relate changes in precipitation depths with the observed changes in streamflow. We do not explicitly evaluate a change in soil moisture.

L185-187 What does it mean for the water balance estimation that a Thiessen polygon results in better matching observations representing the elevation gradient in precipitation? What does this mean for the degree of reality with which the authors represent real-world processes? From the text here, it sounds like the water balance calculation might be quite wrong, and a precipitation distribution that the authors know is not correct has been selected to fit the uncertain water balance calculation. Please add more explanation/justification to show that the Thiessen polygons are still the best representation of precipitation, although they might not reflect the real-world precipitation patterns. A map showing where the two are different and how different they are could be a good start.

Unfortunately, a reliable estimate of net watershed precipitation within this basin would be quite difficult to achieve given the limited availability of data and strong elevation gradients. That the water balance closure is poor using the elevation-gradient suggests that the relationship between precipitation and elevation is nonlinear over the range of elevation in the basin (most likely the relationship flattens at higher elevations). With that said, the water balance closure provides some confidence in the Thiessen polygon interpolation -- it's otherwise impossible to generate an independent measure of the amount of bias in this interpolation. This is a common problem in hydrology and it's why, as we note, some researchers have argued to "leave the water balance open". Our approach relies on the fact that we use the same datasets to estimate precipitation before and after 2000, so that any consistent bias will be removed when calculating this change. Of course, there may be some nonstationary bias that appears or disappears between the two periods (before and after 2000) that we are unable to account for. We again note that our approach hedges against this possibility by using multiple hypothesis testing to corroborate findings and build our understanding of hydrological change. We will nevertheless make this limitation more explicit within the revised manuscript. We will also add the suggested map to show the differences between the two approaches to precipitation interpolation.

L227 how are season defined, and do these match up with crop life cycle? If not, how does that influence the analysis presented?

We use four 3-month seasons commonly defined for Kashmir (e.g., Khattak, 2011). Rice, wheat, and maize are the top three crops in the region. Although the seasons don't perfectly align with the growing season, Summer (June-August) is the primary growing season for paddy and maize. Both are typically planted in the spring and harvested in late summer (August) or early fall (September). Wheat is typically planted in October and harvested in June.

In the spring, the greatest change in NDVI is outside the valley (Figure 5a), which we associate with greater activity of native vegetation. The exception is along the southern portion of the valley, where we observe an increase in fruit orchards. In the summer, the greatest increase in NDVI is seen throughout the valley (Figure 5b), which consists primarily of traditional crops (e.g., paddy and maize) along with the orchards in the south.

We will clarify these details within the revised manuscript.

Khattak, M. S., Babel, M. S., & Sharif, M. (2011). Hydro-meteorological trends in the upper Indus River basin in Pakistan. Climate research, 46(2), 103-119.

L265 I am not convinced that hypothesis 8 or 9 are sufficiently supported with the data as presented now. For instance, quantifying by how much the snowmelt date and peak streamflow date have shifted would give better insight in their relation than just only stating "both were earlier, so the appear to be connected". Hypothesis 9 is not based on independent observations (as shown in Fig 2) but on an inference that because evapotranspiration estimates were higher less water will have recharged the groundwater so that now groundwater contributions are lower. Lastly, the use of 'baseflow' analysis in Figure 2 applicable to hypothesis 9 is confusing, because 'baseflow' analysis is a common term used for the analysis of water level data. In this case, there is some analysis of streamflow data (for which there are no methods - confusing?) but the bulk seems to be based on the extent of surface water, which to me is a less common of the term 'baseflow analysis'. Again, an estimation of the uncertainties that are involved are necessary here.

We will provide the dates of peak snowcover and streamflow before and after 2000, as suggested, and will clarify the limitations of our understanding of the relationship between peak snow cover and streamflow. As for hypothesis 9, we realize that we have not clearly articulated our approach to evaluating this hypothesis. We will first clarify that the hypothesis seeks to identify a reduction of groundwater in the valley. We do this primarily by assuming baseflow can be determined via a (non)linear reservoir model, which is commonly used to associate baseflow ($Q\_B$) with groundwater storage (S) as:

$$Q_B = aS^b.$$

We don't apply or calibrate this model, but rather we use it conceptually to demonstrate that a reduction in baseflow is indicative of a loss of groundwater because there is a monotonic

relationship between the two. By demonstrating that baseflow has declined over time, we infer that there has also been a loss of groundwater. As such the conclusions related to this hypothesis are necessarily circumstantial (we do not have the ability to observe a loss of groundwater). We take care as to how we describe our findings pertaining to this hypothesis. For instance, we state "This lends credence to Hypothesis 9, and we further discuss potential causes and implications of these opposing trends in the baseflow index with respect to saturated and unsaturated groundwater storage in the Discussion (Sect. 5)". This approach is in agreement with the method, which favors holistic understanding through analyses of multiple hypotheses. However interesting, a complete analysis of groundwater is beyond the scope of this manuscript and would be difficult to implement.

L292 remove "by"

We will remove.

L310-317 This section is not clear to me. A drawing would be helpful.

The key idea of this paragraph is that the main river channels are connected to multiple large water bodies, notably Wular Lake and the wetlands in the valley. If, for instance, streamflow below the lak were controlled entirely by lake water levels, we would expect a 1-to-1 relationship between streamflow and the lake water area. Conversely, if streamflow were controlled by a combination of lake water levels and groundwater storage, we would expect the system to exhibit a property of memory based on groundwater storage. Because we expect the river to be coupled with groundwater, we hypothesized that there may be hysteresis in the relationship between water stored in surface water bodies and streamflow at the watershed outlet. We will clarify these points in the manuscript. We include here the original Figure S7 from the supplementary material, which shows this hysteresis:

[Figure]

We will also add the following figure as a subpanel to Figure S7. This figure shows how the observed hysteresis could be produced from the interacting relationships between streamflow, surface water bodies, groundwater, and runoff generation in the upland portion of the watershed. In particular, it could be explained by gaining condition in the summer and losing conditions in other seasons, although testing this follow-up hypothesis requires additional research.

[Figure]

L355 what is the threshold for a 'storm' if storms smaller than 1mm are included? Would the findings also hold if only storms > 5mm are included? (a common threshold)

Our threshold for daily precipitation events is 1 mm / day. The PERSIANN dataset includes data where precipitation is less than 1 mm on a particular day, and we therefore removed such days from our analysis of precipitation events. We will re-run the analysis using a threshold of 5 mm and include our findings in the SI.

L360 32 mm < 15% of 311 mm… please make clear to the reader that the change is smaller than the uncertainty.

Although this statement is true (and we will clarify it in the manuscript), context is required. Even though the 32 mm is less than the "missing" portion of the differential water balance, we have high confidence that there was an increase in ET. We will support this claim in the revised manuscript through a more careful treatment of uncertainty, in response to your second major comment above. In particular, we will compare the change in ET estimated from remote sensing datasets with change in ET estimated from the water balance residual.

L378 please quantify "a substantial level of uncertainty", and the implications that has for this analysis.

Because there are fewer observations, there are greater standard errors and uncertainty in estimating the mean. Although there is high uncertainty for these pixels, they represent a small portion of the watershed. We will clarify the effect of these pixels on the overall estimate of ET in the revised manuscript by showing the fractional areal coverage of these pixels in relation to the watershed.

Fig 6 I spent five minutes looking at this figure, but still don't fully understand what is shown. Moving the legend from panel a outside of the plot region would certainly help, but then still, I am not sure what the individual panels show and how the panels work together.

Thank you for pointing this out. The figure aims to show two pieces of information: (a) the land use categories in which ET increased the most (ie, on a per-unit-area basis), and (b) the land use categories that contributed the most to the change in ET (ie, averaged over the entire watershed). We will seek to address this confusion by:

- Moving the legends outside the plots
- Adding a y-axis title to the plot on the right
- Modifying the caption to: "Changes in land use and evapotranspiration. (a) Change in evapotranspiration within each combination of land use categories from 2001 and 2010. The size of the circle represents the fractional area covered by each pairwise grouping, and the shading of the circle indicates the average change in ET per unit area. For instance, the largest increase in ET occurred in pixels that changed from crops in 2001 to mosaic in 2010 (+80 mm), but this represented a small fraction of the watershed (<5%). (b) Net effect of each 2010 land use category on evapotranspiration, averaged over the entire watershed (i.e., $\Delta$ET * LU Area / Watershed Area). For instance, the pixels that changed from crops to mosaic reduced ET by 2 mm when averaged over the entire watershed. Overall, most of the increase in watershed ET (27 mm) occurred in regions where land cover remained consistent from 2001 to 2010 (outlined in black), compared with regions where land cover changed (5 mm)."

L428 how was the baseflow index calculated? What is the associated uncertainty?

Baseflow was calculated using a numerical filter (Nathan and McMahon, 1990) (we note that this information was in the SI and will be ported to the main text), and the baseflow index was calculated as (baseflow) / (total flow). There are two sources of uncertainty -- first, the uncertainty associated with using trimonthly observations of streamflow, and second, the uncertainty associated with the numerical filter parameter. In the revised SI, we will: (a) benchmark the trimonthly baseflow against baseflow from daily measurements, and (b) conduct a sensitivity analysis on the numerical filter by adjusting the filter parameter within commonly acceptable ranges.

Nathan, R. J. and McMahon, T. A. (1990): Evaluation of Automated Techniques for Baseflow and Recession Analysis, Water Resources Research, 26, 1465–1473, https://doi.org/10.1029/WR026i007p01465.

Fig 8 how was the significance of these analyses tested?

Thank you. We will clarify in the manuscript that the statistical significance in this analysis was determined as p < 0.1 for a nonzero trend from a linear least squares regression. All

statistically significant trends were also significant for p < 0.05 except for the Upstream trend in panel (a).

Fig 9 to me it's not clear what the different subsurface layers mean. Soil? Groundwater? Saprolite?

We will clarify in the revised manuscript that the soil layer in the figure represents unconsolidated deposits in the valley, which can be up to 1 km thick. Near the river channel, these are mostly alluvial deposits which overlay glacial deposits. Further up the slopes (but still in the valley) there are no alluvial deposits and the glacial deposits are uncovered.

L475 Doesn't the conclusion that NDVI was leading for ET follow from a model in which ET is calculated from NDVI? And, why did NDVI increase? The authors describe that warming temperatures were a reason for NDVI to increase, but said that temperature did not influence (evapo)transpiration directly? Precipitation amounts were much lower, and an increase in storms < 1mm don't bring much moisture to the soil either, that will be evaporated directly from the plant leaves. How can the climatic conditions be more favorable, while moisture and temperature apparently don't play a direct role?

We interpret this comment as highlighting the lack of clarity regarding our discussion of changing ET. We will therefore seek to clarify a couple of points. First, Temperature (T) can affect ET directly through the effect on potential evapotranspiration (PET). In the manuscript, we evaluate the effect that temperature has on ET via its effect on PET when we state: "Overall, of the 32 mm annual increase in ET that we detected, approximately 17% can be attributed directly to increasing air temperature." We will clarify that we are specifically referring to the effect through PET. Additionally, temperature can affect ET indirectly by changing growing conditions, vegetation structure and phenology, and therefore NDVI. In many places outside the valley where natural vegetation prevails (e.g., grassland / shrubs), there are large increases in NDVI and ET. This can be seen in Figure 6b by the yellow bar at the top -- land use remained the same but ET nevertheless increased considerably. While we note this relationship, we do not directly evaluate the effect of temperature on NDVI, as it is beyond the scope of this manuscript, and of course NDVI is also dependent on other variables.

Notably, NDVI also depends on land use. In the manuscript, we can make the broad assumption that any changes in NDVI can be attributed to land use change in pixels where land use actually changed. This would give us an upper bound on the effect of changing land use on ET, which we find to be 5 mm. This upper bound is nevertheless small relative to the estimated change in ET of 32 mm, allowing us to conclude that land use change is not a major driver of changing streamflow.

We will clarify the relationship among these variables by adding the following figure:

[Figure]

Caption: Temperature (T) can affect ET directly through the effect on potential evapotranspiration (PET), or indirectly by changing growing conditions, vegetation structure and phenology, and therefore NDVI. Land use also affects NDVI. We evaluate the direct effect of Temperature on ET using the ET model, and the effect of land use change on ET by assuming that any changes in NDVI where land use also changed are attributable only to the change in land use.

S1 pleas add 1:1 lines in the mass-mass plots

Thanks for the comment. We would like to note that double-mass plots should be linear, but not necessarily 1:1 (e.g., if one gauge reports more or less streamflow on average). To address the essence of your comment we will display linear trends on the double mass plots.

S2 what does the overestimation of precipitation say about the accuracy of your input data? What do the authors reckon is the importance of the spatial distribution of rain vs. the basin-average precipitation estimation. Spatial distribution might be very important, which is indicated by the small change in Q for various gauges. Commenting on this would be appreciated.

We don't have a way to validate the uncertainty on precipitation aside from using the water balance to see how far away we are from water balance closure. Because we utilize datasets that span both periods of analysis (before and after 2000), this issue would only affect our analysis if there is a new bias that appears (or disappears) between the periods. Of course, such a change could be plausible if, for instance, there is a change in the altitudinal precipitation gradient. We will therefore clarify in the manuscript that there may be changing biases that we cannot account for, that increase the uncertainty of our analysis (which we now quantify).

S6 what was assumed when zero or only few (defined as less than 5 in the text) images were available?

This comment refers to the number of summer landsat observations (for each pixel) prior to 2000 that were available to estimate NDVI. In some pixels, we had less than 5 cloud-free observations. We do not assume anything for these pixels, but rather note that they may bias the results. In the revised SI, we will provide a histogram of summer landsat observations to demonstrate that the net effect is likely small, because most pixels had a larger number of good observations.

---

## Author Response (AR1)

Dear Dr. Slater,

Please find attached the revised version of our manuscript, previously titled "Empirical attribution of a drying Himalayan river through remote sensing and secondary data", now renamed "Climatic and anthropogenic drivers of a drying Himalayan river", for consideration as a research article at HESS.

We have carefully addressed the concerns raised by two reviewers and are grateful for the opportunity to resubmit. The most substantive changes to the manuscript include:

- Restructuring the introduction to place greater emphasis on the case study. Although we still stress the importance of the methodological approach, we situate the methods relative to the case study, rather than the other way around. This shift in focus prompted us to also change the title.
- Improving the uncertainty analysis and conducting formal hypothesis testing for each of our hypotheses. We also provide additional robustness checks in the supplementary information. Additionally, we have added one hypothesis regarding a change in surface water storage after 2000. This brings the total number of hypotheses to 10 (from 9).
- Clarifying a variety of methodological details including adding a new Figure (Figure 3) that shows how temperature and land use each affect evapotranspiration.

We are grateful to both reviewers and to yourself for taking the time to consider our paper and for providing such constructive comments. We believe that the revised manuscript has been substantially improved thanks to this feedback and hope that we have addressed all of the reviewers' concerns. A point-by-point response (in blue) to all reviewers' comments and the related revised text (in magenta) are provided below. A marked-up version of the manuscript (with tracked changes) and revised SI is enclosed.

Thanks again for your consideration.

Sincerely,

Gopal Penny, on behalf of all authors.

Comments from Editor:

Dear Authors,

Thank you for your detailed responses to the two referees who reviewed your manuscript. Both reviewers find that the manuscript presents an interesting 'multiple hypotheses' approach to understanding the potential drivers of changing streamflow in the Upper Jhelum river region. They both however raise some issues and make some very relevant suggestions regarding the framing of the manuscript and relation to prior studies, clarification of the methods, and estimation and clarification of uncertainties. Based on your responses to the referees, I would like to invite you to submit a revised manuscript fully addressing these concerns. The manuscript will then be re-reviewed by myself and the same reviewers. I look forward to reading your revised paper.

Sincerely,

Louise Slater

Thank you for the comments and suggestions. We wish to note that we have considered the suggestion to include ERA5-Land data in our analysis, in particular to supplement our analyses with soil moisture data, but we are hesitant to use reanalysis data for soil moisture given the difficulty in untangling the effects of multiple drivers of soil moisture (total precipitation, distribution of precipitation events and arrival times, seepage, etc) in addition to possible discrepancies between the model and observations. On the other hand, we have included in the Supplementary Information a map of the changes (pre/post 2000) in monthly snowmelt from ERA5-Land, as the snowmelt signal is likely dominated by the effect of warming temperature, and helps to address comment #10 of Reviewer 2. With that said, the main text focuses purely on empirical data (remote sensing products or gauge data), including evaluation of each of the hypotheses.

**Response to Reviewer 1**

Reviewer comment: https://doi.org/10.5194/hess-2021-274-RC1

I read this paper with interest. It deals with an issue that is becoming more and more important, i.e., hydrologic change. Increasingly we are having to attribute possible causes of change and to develop models that can reproduce past change and predict change in the future. In this paper the authors are dealing with the Upper Jhelum catchment in the headwaters of the Indus River, which has experienced decreasing streamflows. The paper is aimed at attributing possible causes of the decreasing streamflow. They implement the notion of multiple working hypotheses to assess a combination of alternative causal explanations for this observed decrease. I found the analyses to be insightful and valuable, albeit somewhat inconclusive due to data limitations and uncertainty.

While the paper is publishable (eventually), I found the presentation to be confusing and unnecessarily diffuse.

We would like to thank the reviewer for considering the manuscript and providing constructive and helpful feedback. In particular, we appreciate the suggestion to re-frame the manuscript as a case study and hope that this has brought clarity to the motivation (both for the manuscript and the need for the particular methods we use). As the following three comments relate to the framing of the paper, we address them together, below.

1. Firstly, for some reason, the authors have conceived the paper as a methodological advance, and have gone to a lot of trouble to give an overly philosophical introduction, talking about top-down and bottom-up and multiple working hypotheses etc. I had to read four manuscript pages before they introduced the problem they are studying. I would like them to start with data evidence of both the cause and effect, before presenting the alternative hypotheses as part of their methodological presentation.

2. Secondly, this is a place-based study, the sooner you get to the problem, the better its readability. The philosophical discussion can be kept to a minimum. I would like to know more about the streamflow decrease as early as possible, and the history of climatic, and anthropogenic changes to land use and land cover as possible causal factors. In other words, I like more of an introduction to the basin and its hydrology that can help motivate why it is that they form their multiple hypotheses.

3. Thirdly, I am not even sure if the authors are correct in calling their modeling approach a bottom-up study. The way I read it, the authors analyze data at the catchment scale and find a decreasing pattern of streamflow. They then systematically try to attribute it to change in rainfall, evapotranspiration, land use and land cover change etc, or a combination of these. In my opinion, this is in fact a data-based, top-down study. I go back to the original source (Klemes, 1983), who defines the downward (top-down) approach "starts with trying to find a distinct conceptual node directly at the level of interest (or higher) and looks for the steps

that could have led to it from a lower level".  This is the way I see the authors' attribution work.

Thank you for these comments. We agree that the paper focuses primarily on understanding the hydrology of the Upper Jhelum. We have therefore increased the emphasis of the Upper Jhelum within the introduction, and reduced the emphasis on the methodological approach. With that said, we do not want to relegate the approach entirely to the methods section, because we believe it presents an alternative to the many emerging studies of hydrological attribution that rely on predictive inference (e.g., basin hydrological simulation), particularly when focusing on watersheds at spatial scales as large as the Upper Jhelum. We had hoped to emphasize this contrast by referring to our study as a "bottom-up" attribution. However, we now recognize that this terminology is confusing and we now avoid it.

In response to these comments, we have restructured the Introduction so that the Upper Jhelum is introduced in the second and third paragraphs. The remaining components of the introduction with respect to attribution (and associated methods) has been reduced and serve in support of the goals of the study site. In providing greater emphasis on the Upper Jhelum, we also discuss the decline in streamflow in greater detail and more clearly review potential causes and mechanisms for the drying of the upper Jhelum.  We hope this provides a better framing for our approach, hypotheses, and subsequent analysis.

4. Finally I would like to see a clear statement of their conclusions, focusing more on the hydrology of the place and perhaps less on methodological sophistication. In my opinion, the paper raises a number of important issues about the hydrology of this region, and placing the result in the context of the local hydrology would be much more useful than whether this is a bottom-up or top-down method.

Per your suggestion, we have de-emphasized the methodological components throughout the paper (including removing references to bottom-up and top-down) and focus the paper more on the implications for local hydrology. Additionally, in the conclusions, we better situate the methodological discussion as part of a broader discussion on the hydrology of the Upper Jhelum, including key questions for future research.

5. The paper is definitely publishable after major revisions to address or discuss (in case the authors want to dispute my comments) the issues I have raised in my review

We thank the review for constructive comments and believe our revisions have considerably strengthened the manuscript.

**Response to Reviewer 2**

Reviewer comment: https://doi.org/10.5194/hess-2021-274-RC2

The Penny et al. manuscript presents a variety of observations to investigate why the water balance for a Himalayan river has changed. They highlight the limits to a simple water balance calculation and instead include information on land-use, NDVI and water extent as markers for differences in vegetation or crop productivity (i.e., evapotranspiration) and groundwater contributions to surface water, respectively. They focus on the changes between a 15-year (1984-1999) and a 13-year (2000-2013) period, and approach the work as a 'method of multiple hypotheses', testing various hypotheses related to potential drivers of the change in streamflow. I found the manuscript relatively well-structured and I think that it can be a good addition to the literature and a good resource for water managers in the Jhelum river region. However, I do think the manuscript needs some work before it is ready for final publication.

Thank you for the careful consideration and constructive feedback, which we believe strongly improve the manuscript. We respond to each of your comments below. We have added numbers to make it easier to refer to specific comments.

1. From a clarity perspective, I found that some words were not well defined ("secondary data", "drivers", "changing climate") and that the reader would benefit from describing these better. From a structural perspective, I found that the methods and results were not always matching up. How was the baseflow index calculated? How is a storm event characterized? Why is a basin-average correct value for precipitation preferred over a locally correct value? How were statistical analyses performed?

In the revised manuscript, we have clarified each of these terms early in the introduction. The approach for calculating for baseflow was previously relegated to the SI, and we haved ported this to the manuscript and provided additional details (as described below). The remaining questions were addressed by expanding the methods sections. These issues are raised in the Detailed comments below, and we address them individually under each comment.

With respect to defining "drivers", we now write (L24):

In order to address the management challenges of mitigation and adaptation, observed changes in the hydrological processes must be correctly attributed to the corresponding drivers (i.e, the processes that cause changes in hydrology).

Our use of "secondary data" is clarified below, in response to comment #3. We have removed any uses of the phrase "changing climate". Where a replacement was needed, we used more specific terms (e.g., "temperature increases").

2. What was also clearly missing in the manuscript were uncertainty estimations on the data used to test the various hypotheses. The authors indicate an uncertainty in their initial water balance estimation of 15% (which by the way is more than increase in ET and almost as much as the decrease in streamflow of 117 mm?). An estimation of the uncertainty related to individual components of the water balance would be helpful here – or is it 15% of each component? Uncertainty estimations were given for none of the following analyses… (precipitation estimations from remotely sensed observations, evapotranspiration estimates using crop factors, NDVI estimations when only few (=< 5) images were available, and inferring catchment storage from surface water extent). I expect there to be considerable uncertainty in each of these estimations, and expect that to be quantified and discussed in the text of the updated manuscript.

Thank you for this comment. Estimating uncertainty represents an important challenge in this study. Before describing how we address this comment, we wish to emphasize that the manuscript is oriented towards improving conceptual understanding of hydrological change. A key objective of the paper is to demonstrate the use of multiple hypotheses as an alternative approach for attribution in situations where data scarcity and uncertainty confounds the application of a model-based predictive inference approach. One important reason we included the water balance was to give a sense of the overall uncertainty with respect to water balance fluxes and therefore inform interpretation of the overall approach -- the water balance remains a key aspect for interpreting uncertainty, and we expand on this as described below. With these points in mind, we completely agree that observation uncertainties should not only be used to justify our approach, but also systematically incorporated into the approach itself.

We have improved our analysis of uncertainty in three ways, with an emphasis on providing a stronger statistical foundation for the evaluation of the hypotheses through (1) formal hypothesis testing, (2) comparison of precipitation interpolation, and (3) water balance residuals.

1. Hypothesis testing

We have run statistical tests for hypotheses 1-5 and 8-10, by bootstrapping 95% confidence intervals and determining statistical significance in cases where the confidence interval excludes zero, noting that we have added Hypothesis 10 to formally test for changes in surface water. Each of these tests is now described in the methods section, and we believe give greater confidence in our evaluation of each individual hypothesis. Hypotheses 6 and 7 deal with glacier and permafrost contributions to the change in streamflow. For these hypotheses we only have bounded estimates of change. Instead of traditional hypothesis testing we provide upper-bound estimates on the expected changes. The results of these tests are now shown in Table 2 (see below).

2. Precipitation interpolation

We now better describe uncertainty due to spatial variability of precipitation by showing maps of the precipitation interpolation highlighting the differences between the two interpolation methods (Figure S5). We also better describe how using the different regionalization approaches would affect our hypotheses. In particular, although the elevation-gradient interpolation yields a much larger estimate of annual precipitation and a greater value of change (pre- versus post-2000), the results of the before-after comparison do not alter the main conclusions with regards to precipitation (see Figure S6 below), which is identical to Figure 4 (shown below) except that we used precipitation from the elevation-gradient approach rather than Thiessen polygons.

3. Water balance residual

As an independent estimate of the uncertainty associated with each component of the water balance, we now compare our findings to what would be expected from a water balance closure. The results are presented in the revised Figure 4 (see below).

Ultimately, the benefit of the multiple hypotheses approach is that the findings from some hypotheses can corroborate or cast doubt on the findings from other hypotheses. The fact that the findings from many of the hypotheses lead to conclusions that are consistent gives credence to the findings that we present.  With that said, we agree that a more careful treatment of uncertainty not only provides greater confidence in the approach but greater context for readers interested in the Upper Jhelum, and we are thankful to the reviewer for this suggestion.

For reference, we include Table 2 of the main text which shows the results of hypothesis testing for each hypothesis:

Table 2. Summary of hypothesis tests, with 95% bootstrapped confidence intervals. Statistical significance occurs when the confidence interval excludes zero.

| Hyp. | Change in: | Conf. int | Units | Sig |
|---|---|---|---|---|
| 1 | Total precipitation | [-172.1, -107.2] | mm | Y |
| 2 | Winter precipitation | [ -61.5, 61.3] | mm | |
| | Spring precipitation | [-185.3, -44.8] | mm | Y |
| | Summer precipitation | [ -57.6, 30.7] | mm | |
| | Autumn precipitation | [ -37.9, 29.8] | mm | |
| 3 | Number of large events | [ -3.8, -2.5] | - | Y |
| | Number of medium events | [ -0.5, 1.6] | - | |
| | Number of small events | [ 14.1, 18.0] | - | Y |
| 4 | Temperature effect on Winter ET | [ -0.9, 1.3] | mm | |
| | Temperature effect on Spring ET | [ 1.2, 5.2] | mm | Y |
| | Temperature effect on Summer ET | [ -2.3, 1.9] | mm | |
| | Temperature effect on Autumn ET | [ -0.1, 2.3] | mm | |
| 5 | NDVI effect on Winter ET | [ -0.6, -0.3] | mm | Y |
| | NDVI effect on Spring ET | [ 7.0, 7.5] | mm | Y |
| | NDVI effect on Summer ET | [ 12.7, 13.7] | mm | Y |
| | NDVI effect on Autumn ET | [ 2.8, 3.2] | mm | Y |
| 6 | Glacier melt (upper bound) | [ NA, 0.0] | mm | |
| 7 | Permafrost melt (upper bound) | [ NA, 0.6] | mm | |
| 8 | Peak snow cover date | [ -27.0, -5.0] | days | Y |
| | Peak streamflow date | [ -28.0, 0.0] | days | Y |
| 9 | Annual baseflow | [-436.8, -241.7] | mm | Y |
| 10 | Wular lake extent | [ -24.1, -8.2] | $km^2$ | Y |
| | Valley inundation | [ -14.0, -3.0] | $km^2$ | Y |

We have added the following figure and caption to the SI, showing the spatial difference in interpolation approaches:

[Figure]

Figure S5: Precipitation interpolation using Thiessen polygons (left) and inverse distance squared with an elevation gradient (right), including (a, b) Pre-2000, (c, d), Post-2000, and (e, f) the change between the two periods. Given the difficulty of measuring precipitation in regions with strong elevation gradients and significant snowfall, we used a water balance closure to interpret the validity of the two methods. In both periods (pre- and post-2000), the Thiessen polygon approach yields less uncertainty in the water balance closure.

We also include the revised Figure 4:

[Figure]

Figure 4. Water balance for the Upper Jhelum. (a) Average annual fluxes (P, Q, and ET) for the periods 1984–1999 and 2000-2013. (b) Average annual change in water balance fluxes between the two periods, given by the bars, with each flux estimated independently. The change in water balance fluxes estimated from water balance residuals is given by the arrows. In other words, each flux is calculated using the equation $\Delta P - \Delta Q - \Delta ET = 0$ and estimates of the other two fluxes.

We also include the revised Figure S6:

[Figure]

Detailed comments:

3. Title: What do the authors mean with 'secondary data'? – being specific would be helpful here to guide the potential reader to reading the article.

First, we note that we removed "secondary data" from the title, which has been revised to: "Climatic and anthropogenic drivers of a drying Himalayan river" -- a change made to reduce emphasis on the methods. Second, we have clarified that 'secondary data' refers to datasets collected by other sources -- in this case, mostly governmental agencies. The reason the emphasis on secondary data is important is because collecting primary hydrological data in remote (or transboundary) locations is often impractical, and therefore reliable approaches that build on publicly available data sources can more easily support attribution. We have clarified within the introduction the definition of secondary data on its first usage (L113):

... particularly when combined with secondary data(i.e., historical data collected by third parties, such as government agencies).

4. L28 Instead of pointing out that there are many watersheds where the hydrological drivers are unknown, which is not surprising given the amount of watersheds globally, highlighting prior studies that did identify and quantify drivers would be more helpful.

Thank you -- in response to this comment and comments by the other reviewer, we have condensed this part of the introduction and removed the sentence in question in favor of emphasizing the case study and introducing it earlier. The introduction has therefore been restructured so that the majority of the literature review (and description of our approach) serves the case study, rather than the methods. With respect to the suggested literature review, we have sought to highlight existing attribution studies by the different approaches that have been used, in order to describe how those approaches are unlikely to be appropriate for the Upper Jhelum. This passage is on L57-81 of the revised manuscript.

5. L34 "Their associated drivers" which drivers are meant here?

We clarify that by "associated drivers" we mean, broadly, the "causal drivers of hydrological change". Please see the revised text in response to #1.

6. L76 "finding"s please add the s

This sentence has been removed.

7. L78 remove "by"

This sentence has also been removed.

8. L124-129 consider removing this text since a standard manuscript format is followed, or adapt such that it contains information that is specific to this manuscript.

Thank you -- we have removed this paragraph.

9. Fig 1 The panels 1d-g are square, but they are not square where they are indicated on map 1b. Did the coordinate system change? Then please indicate the new coordinates along insets d-g. If not, please make sure that the indication and maps match. Also, for me it would have been helpful to have the line of the river drawn in panels d-g, and the river and catchment boundary are not shown in the legend (I assume light blue and black lines).

We have fixed the projection in this figure so that the shapes of the subpanels are consistent and we have also included the main river channels.

10. Fig 2 snow density is needed to calculate water content from area and depth, and should be included in this scheme as well as in the calculation.

We now clarify in the text that we do not calculate snow storage volumes, but rather focus on snow areal coverage. This is consistent with the revised hypothesis 8, which states that "Earlier snowmelt contributed to an earlier peak in the annual hydrograph." We now estimate the dates of peak snow cover and streamflow and provide confidence intervals for the change in both. Both peak snow cover and peak streamflow occur earlier after 2000, and the changes were statistically significant in both cases. We therefore accept the hypothesis. With respect to snow storage volumes, we have included in the SI results from the ERA5-Land reanalysis dataset showing monthly changes in snowmelt before and after 2000 (Figure S13). We have not included these results in the main text (or for hypothesis testing) because we wished to keep with our original approach of using empirical data wherever possible, rather than simulated results.

11. L173-174 define "changing climate" and justify how a dataset that is shorter than 30 years can be representative of a change in climate.

We agree that a change in precipitation or temperature over a 3-decade period may or may not be associated with climate change. We hope the revised text avoids this confusion as we have updated these hypotheses to:

- Hypothesis 1: Annual precipitation declined.
- Hypothesis 2: Climate exhibited a change in rainfall seasonality

12. L178 'a reduction in storm size' is not captured in the hypotheses yet – greater storm frequency does not always mean in a reduction in storm size. Also, there was no explanation on how 'storms' were characterized (precipitation magnitudes separated by an interval of how many hours at least?). Lastly, in many parts of the world a 'storm' can just be

a period with a lot of wind, and does not necessarily imply rainfall. Please consider changing it to 'rainfall event' or 'precipitation event'.

Thanks for this comment. We now avoid the use of "storms" in favor of "precipitation events" in the revised manuscript. These events correspond to days with precipitation, in accordance with the fact that we only have access to daily precipitation via PERSIANN. We have revised the hypothesis to focus on the frequency of events of different sizes by grouping precipitation events into small, medium, and large events. We re-ran the analysis using different bin sizes to ensure that bin width was not the determining factor in our findings. The manuscript text now reads (L171-176):

Hypothesis 3: The distribution of precipitation events changed.

A shift in the distribution of rainfall events could have various effects. A shift towards smaller event sizes would likely reduce the "fast" component of streamflow (i.e., quickflow, McCaig, 1983) and leave a greater fraction that is directly intercepted, re-evaporated, or infiltrated. In other words, a shift in precipitation patterns towards more frequent small events and fewer large events could lead to a reduction in runoff and increase in evapotranspiration (Zhao et al., 2019). These changes would reduce quickflow and groundwater recharge, and thus ultimately reduce annual streamflow volume.

We describe the approach on L184-193:

The monthly frequency of the IMD precipitation dataset precluded analysis of the distribution of precipitation event sizes. Therefore, we utilized daily precipitation records from the gridded PERSIANN Climate Data Record (CDR, Ashouri et al., 2015). The purpose of PERSIANN CDR is to provide consistent precipitation data for long-term climate analysis dating back to 1984 at 0.25 degree resolution. Consistent with the daily frequency of PERSIANN rainfall data, we here define a precipitation event as any day with precipitation ≥2 mm. In order to determine a shift in the distribution of event sizes, we binned events into three groups: small (2–7.4 mm), medium (7.4–18.6 mm), and large (>18.6 mm) events. The size of the bins was determined such that the total precipitation from all events within each bin was equivalent (i.e., the sum of precipitation from all events in the small bin was equivalent to the sum in the medium and large bins). We then determined whether or not the number of precipitation events in each bin changed before and after 2000.

As noted above, we re-ran the analysis with different bins. The results using a different minimum event size (which also results in different bins) are included in the SI:

Table S1: Robustness check for a change in the distribution of precipitation event sizes. The same analyses were run as described in the main text (Sect. 3.2.2), with minimum storm sizes of 1 mm and 3 mm. The event size bins for 1 mm were: small (1–6.4 mm), medium (6.4–17.4 mm), and large (≥17.4 mm) events. The event size bins for 3 mm were: small (3–8.4 mm), medium (8.4–20.1 mm), and large (≥20.1 mm) events.

| Hyp. | Change in: | Min event size | Conf. int | Sig |
|---|---|---|---|---|
| 3 | Number of large events | 1 mm | [-4.1, -2.7] | Y |
| | Number of medium events | 1 mm | [-0.2, 2.2] | |
| | Number of small events | 1 mm | [31.2, 36.7] | Y |
| | Number of large events | 3 mm | [-3.6, -2.4] | Y |
| | Number of medium events | 3 mm | [-0.8, 1.1] | |
| | Number of small events | 3 mm | [ 6.0, 9.1] | Y |

13. L181 The authors hypothesize here about greater bare soil evaporation, but included no information on soil moisture (only groundwater), neither did they explain why this information is not considered.

We are unaware of any empirical data source that would provide reliable information about soil moisture before and after 2000, and therefore we do not explicitly evaluate a change in soil moisture. For this reason we can only hypothesize what could have changed. To avoid confusion, we have revised the hypothesis to focus on the distribution of rainfall event sizes and we have removed the reference to soil moisture to focus more directly on the effect of changing event sizes on the water balance. We have included the revised text for this hypothesis in response to comment #12.

14. L185-187 What does it mean for the water balance estimation that a Thiessen polygon results in better matching observations representing the elevation gradient in precipitation? What does this mean for the degree of reality with which the authors represent real-world processes? From the text here, it sounds like the water balance calculation might be quite wrong, and a precipitation distribution that the authors know is not correct has been selected to fit the uncertain water balance calculation. Please add more explanation/justification to show that the Thiessen polygons are still the best representation of precipitation, although they might not reflect the real-world precipitation patterns. A map showing where the two are different and how different they are could be a good start.

Unfortunately, a reliable estimate of net watershed precipitation within this basin would be quite difficult to achieve given the limited availability of data and strong elevation gradients. That the water balance closure is poor using the elevation-gradient suggests that the relationship between precipitation and elevation is nonlinear over the range of elevation in the basin (most likely the relationship flattens at higher elevations). With that said, the water balance closure provides some confidence in the Thiessen polygon interpolation -- it's otherwise impossible to generate an independent measure of the amount of bias in this interpolation. This is a common problem in hydrology and it's why, as we note, some researchers have argued to leave the water balance open (i.e., "The case for an open water

balance," Kampf et al., 2020). Our approach relies on the fact that we use the same datasets to estimate precipitation before and after 2000, so that any consistent bias will be removed when calculating this change. Of course, there may be some nonstationary bias that appears or disappears between the two periods (before and after 2000) that we are unable to account for. We again note that our approach hedges against this possibility by using multiple hypothesis testing to corroborate findings and build our understanding of hydrological change. We have modified the introduction to clarify this point (see text below).

Additionally, the two approaches to precipitation interpolation yield similar results in terms of water balance closer on the change between the pre-2000 and post-2000 periods (see Fig S6 in response to comment #2, and note that the residual errors represented by the arrows in panel (b) are similar to those from the Thiessen polygon interpolation in Figure 4). We have also added the suggested map to the SI (see Figure S5 in response to comment #2) to show the differences between the two approaches to precipitation interpolation.

The revised introductory text reads (L91-97).

Second, the approach mitigates against observational and conceptual biases in which certain hypotheses are favored based on preconceived notions of change (Railsback et al., 1990). Using consistent observational datasets before and after 2000 ensures that only a nonstationary bias would affect the results of the analyses. Further, by employing separate analyses for each hypothesis, we construct a process-based narrative of attribution in which each analysis comprises part of the whole and serves to corroborate or contradict other analyses. Indeed, the method of multiple hypotheses advocates broader understanding of the whole over in-depth understanding of individual components. By favoring breadth over depth we seek to develop a coherent and holistic narrative of change.

15. L227 how are season defined, and do these match up with crop life cycle? If not, how does that influence the analysis presented?

We use four 3-month seasons commonly defined for Kashmir (e.g., Mahmood et al, 2011). Rice, wheat, and maize are the top three crops in the region. Although the seasons don't perfectly align with the growing season, summer is the primary growing season for paddy and maize, and winter is the primary growing season for wheat.

In the spring, the greatest change in NDVI is outside the valley (Figure 5a), which we associate with greater activity of native vegetation. The exception is along the southern portion of the valley, where we observe an increase in fruit orchards. In the summer, the greatest increase in NDVI is seen throughout the valley (Figure 5b), which consists primarily of traditional crops (e.g., paddy and maize) along with the orchards in the south.

We have clarified the seasons and growing seasons within the revised manuscript. We now state on L125:

The climate in Kashmir is characterized by four seasons (Mahmood et al., 2015) including winter (December–February), spring (March–May), summer (June–August), and autumn (September–November).

We now state on L136:

Summer is the primary growing season for paddy and maize, which are sowed in the spring and harvested in the late summer or early autumn. Wheat is typically planted in October and harvested the following June.

And we have added, on L443-445:

The most dramatic increases in ET occurred within agricultural land cover classes (cropland and mosaic vegetation, i.e. orchards), which constituted 27% of the catchment area in 2001. In these classes, NDVI increased substantially in the valley in the spring and summer seasons (Fig. 6ab), corresponding with the primary growing season for paddy, maize, and orchards.

16. L265 I am not convinced that hypothesis 8 or 9 are sufficiently supported with the data as presented now. For instance, quantifying by how much the snowmelt date and peak streamflow date have shifted would give better insight in their relation than just only stating "both were earlier, so the appear to be connected". Hypothesis 9 is not based on independent observations (as shown in Fig 2) but on an inference that because evapotranspiration estimates were higher less water will have recharged the groundwater so that now groundwater contributions are lower. Lastly, the use of 'baseflow' analysis in Figure 2 applicable to hypothesis 9 is confusing, because 'baseflow' analysis is a common term used for the analysis of water level data. In this case, there is some analysis of streamflow data (for which there are no methods - confusing?) but the bulk seems to be based on the extent of surface water, which to me is a less common of the term 'baseflow analysis'. Again, an estimation of the uncertainties that are involved are necessary here.

Thank you for these suggestions which have helped us to improve the analyses and clarify the manuscript.

With respect to hypothesis 8, we have refined our analyses by including formal hypothesis testing by bootstrapping confidence intervals on the dates of peak snow cover and the dates of peak streamflow before and after 2000. Both the date of peak snow cover and peak streamflow occur earlier after 2000 (ie, a lower day-of-the-year), supporting the hypothesis. We also note that we have refined the hypothesis, as shown below, and now describe the bootstrapping approach in the methods and clarify the findings with respect to the hypothesis. The manuscript additions are included below.

As for hypothesis 9, we realize that we have not clearly articulated our approach to evaluating this hypothesis. We first clarify that the hypothesis seeks to identify a reduction

of groundwater in the valley by detecting a reduction in baseflow. We do this primarily by assuming baseflow can be determined via a (non)linear reservoir model, and we have clarified this in the manuscript (see the revised manuscript text, below). By demonstrating that baseflow has declined over time (with results of hypothesis testing presented in Table 2), we infer that there has also been a loss of groundwater. As such the conclusions related to this hypothesis are necessarily circumstantial (we do not have the ability to observe a loss of groundwater), we take care as to how we describe our findings pertaining to this hypothesis.

Hypothesis 8 has been revised to (L283):

 Earlier snowmelt  **contributed to** an earlier peak in the annual hydrograph.

The findings with respect to Hypothesis 8 are included on L481-487 of the revised manuscript:

We found statistically significant changes in the seasonal timing of peak snow cover and peak streamflow. After 2000, peak snow cover occurred 15.4 days earlier and the confidence interval included zero. Peak streamflow occurred 13.2 days earlier on average, and the upper limit confidence interval was equal to zero. This provides confidence that the earlier peak in snowmelt led to an earlier peak in streamflow, confirming Hypothesis 8. We note that observational data was missing to quantify the effect of changes in snow storage volumes. However, reanalysis data from ERA5-Land indicates that snowmelt increased in earlier months and decreased in later months after 2000, at all elevations in the watershed (see Fig. S13). Data from MODIS also shows an earlier peak in snow cover after 2000 (Fig. S14).

For hypothesis 9, the revised manuscript now describes the association between baseflow and storage and associated methods for baseflow separation (L351-367):

...we conceptualize the catchment as a (potentially non-linear) 'bucket' reservoir, where baseflow discharge $Q_B$ is positively related with mobile groundwater storage S:

$Q_B = a\ S^b$,

where $a$ and $b$ are positive recession coefficients. Under these conditions, a reduction in baseflow can be considered indicative of a reduction in groundwater storage.

To evaluate whether baseflow declined, we apply a recursive digital filter (Nathan and McMahon, 1990) to the streamflow data to separate quickflow and baseflow (see Figure S3). We used two decades of daily flow at Baramulla station (the most downstream station) to calibrate the filter, which requires a single smoothing parameter, α. We first set α = 0.925 for daily flow based on analysis from Nathan and McMahon (1990), and then tuned

the value of alpha to 0.45 for three-monthly flow so that the annual baseflow index (QB/Qtotal) matched that of daily flow (Figure S4). Lastly, to ensure the results were not sensitive to the specific selection of α, we re-ran the analysis with α = 0.4 and α = 0.5. Although the baseflow index changed, the changes were immaterial for the before-after comparisons as the results were highly correlated across combinations of all three cases ($R^2 > 0.999$).

To formally evaluate Hypothesis 9, we bootstrapped confidence intervals (N = $10^4$) to determine if there was a decline in annual baseflow in the valley in the periods before and after 2000. We also used linear regressions on quickflow, baseflow, and the baseflow index to better understand the causes of the changes in baseflow.

17. L292 remove "by"

Done.

18. L310-317 This section is not clear to me. A drawing would be helpful.

The key idea of this paragraph is that the main river channels are connected to multiple large water bodies, notably Wular Lake and the wetlands in the valley. If, for instance, streamflow below the lake were controlled entirely by lake water levels, we would expect a 1-to-1 relationship between streamflow and the lake water area. Conversely, if streamflow were controlled by a combination of lake water levels and groundwater storage, we would expect the system to exhibit a property of memory based on groundwater storage. We hypothesized that there would be hysteresis in the relationship between water stored in surface water bodies and streamflow at the watershed outlet, and that this relationship would corroborate the influence of a fluctuating water table on streamflow. We have clarified these points in the manuscript on L368-375:

Additionally, to better assess our assumptions, we looked for a hysteresis in the relationship between surface water storage and streamflow. A clear hysteresis would build confidence that there is a connection between subsurface storage and streamflow, because this connection can only occur if groundwater levels are sufficiently high to reach the elevation of the lake and wetlands. A weak or non-existent hysteresis might be indicative of a weak influence of groundwater on streamflow. In contrast, a strong hysteresis would indicate both a greater influence of groundwater and the occurrence of meaningful fluctuations in the water table. Comparison between surface water extent (bothWular lake and valley inundation) and streamflow reveals a clear seasonal hysteresis (Fig. S11).We interpret these results as an indication that groundwater plays an important role mediating streamflow generation in the valley bottom.

We also include here the revised Figure S11 from the supplementary material. The top panel (a and b, included in the original SI, shows this hysteresis. The bottom panel (c) is new and shows how the observed hysteresis could be produced from the interacting

relationships between streamflow, surface water bodies, groundwater, and runoff generation in the upland portion of the watershed. In particular, it could be explained by gaining condition in the summer and losing conditions in other seasons, although testing this follow-up hypothesis requires additional research.

[Figure]

19. L355 what is the threshold for a 'storm' if storms smaller than 1mm are included? Would the findings also hold if only storms > 5mm are included? (a common threshold)

In order to distinguish the effects of different storm sizes, we have re-run this analysis to focus on changes in small (2–7.4 mm), medium (7.4–18.6 mm), and large (≥18.6 mm) events. By binning precipitation this way, we distinguish the effect on different events sizes. The bins are set up such that the total precipitation that arrives in small events is equal to the total precipitation that arrives in medium events and large events, respectively. This means that by changing the minimum event size, the bin widths also change. To ensure that our analysis was robust to changes in the minimum event size, we re-ran the analysis with a minimum storm size of 1 mm and 3mm. Please refer to our response to comment #12 for the results.

20. L360 32 mm < 15% of 311 mm… please make clear to the reader that the change is smaller than the uncertainty.

This statement is true, but requires appropriate context -- which is that the results are statistically significant and in agreement the the results of the water balance closure. We have added this information to the revised manuscript (L436-438):

Annual average watershed evapotranspiration increased by 32 mm, from 311 mm before 2000 to 343 mm after 2000. Although this change is smaller than the uncertainty in water balance closure (i.e., 15% of precipitation), the results are statistically significant and the direction is in agreement with the estimate of ET from water balance residual.

21. L378 please quantify "a substantial level of uncertainty", and the implications that has for this analysis.

The original manuscript text refers to the fact that some pixels were poorly represented by landsat summer images before 2000 (ie, ≤5 images). Although there is greater uncertainty for these pixels than others with more than 5 images, these pixels represent a small portion of the watershed. We have clarified the effect of these pixels on the overall estimate of ET in the revised manuscript by showing (a) that the fractional areal coverage of these pixels in relation to the watershed is small (14%) and (b) there appears to be a downward bias on the effect of the change in NDVI in these pixels, meaning that the outcome of this bias is that the results of the hypothesis test are more conservative.

We state in the manuscript (L456):

In these pixels, few (≤5) summer NDVI observations are available before 2000 due to cloud cover, and these pixels exhibit a downward bias in the reported NDVI change relative to pixels with more images (see Fig. S8). This suggests that any potential bias would lead to a conservative estimate on the change in ET.

Here we include the new Figure S8, which contains this information:

[Figure]

Figure S8: Histogram of pre-2000 clear-sky summer images at each pixel (top). NDVI change (from before to after 2000) versus the number of clear-sky summer images in each pixel (bottom). As shown by the moving (LOESS) regression, NDVI was insensitive to the number of clear-sky images except when the number of images was less than or equal to 5. These pixels account for 14% of the watershed, and therefore have a small effect on watershed-average NDVI.

22. Fig 6 I spent five minutes looking at this figure, but still don't fully understand what is shown. Moving the legend from panel a outside of the plot region would certainly help, but then still, I am not sure what the individual panels show and how the panels work together.

Thank you for pointing this out. The figure aims to show two pieces of information: (a) the land use categories in which ET increased the most (ie, on a per-unit-area basis), and (b) the land use categories that contributed the most to the change in ET (ie, averaged over the entire watershed). We have sought to address this confusion by:

- Moving the legend on the left outside the plot
- Modifying the caption and providing examples. The revised figure and caption are:

[Figure]

"Changes in land use and evapotranspiration. (a) Change in evapotranspiration within each combination of land use categories from 2001 and 2010. The size of the circle represents the fractional area covered by each pairwise grouping, and the shading of the circle indicates the average change in ET per unit area. For instance, the largest increase in ET occurred in pixels that changed from crops in 2001 to mosaic in 2010 (+80 mm), but this represented a small fraction of the watershed (<5%). ET increased in every category. (b) The cumulative change in ET by 2010 land use category (note that the y-axis continues across both panels). The cumulative effect (or watershed-average) ET is equal to the Change in ET × Fraction of watershed for each land use. For instance, the pixels that changed from Crops in 2001 to Mosaic in 2010 reduced ET by 1.4 mm when averaged over the entire watershed. Overall, most of the increase in watershed ET (27 mm) occurred in regions where land cover remained consistent from 2001 to 2010 (b, see bars outlined in black), compared with regions where land cover changed (5 mm, no outline)."

23. L428 how was the baseflow index calculated? What is the associated uncertainty?

Baseflow was calculated using a recursive digital filter (Nathan and McMahon, 1990) and the baseflow index was calculated as (baseflow) / (total flow). In addition to porting this information from the SI, we had added additional clarification about how baseflow was calculated from three-monthly flow. There are two sources of uncertainty -- first, the uncertainty associated with using trimonthly observations to calculate baseflow, and second, the uncertainty associated with the numerical filter parameter. In the revised manuscript, we now: (a) benchmark the trimonthly baseflow against baseflow from daily

measurements, and (b) conduct a sensitivity analysis on the numerical filter by adjusting the filter parameter within commonly acceptable ranges.

These issues are clarified in the revised manuscript, and the new text is presented in response to comment #16.

24. Fig 8 how was the significance of these analyses tested?

Thank you. We now clarify in the figure caption:

The statistical significance in this analysis was determined as p < 0.1 for a nonzero trend from a linear least squares regression. All statistically significant trends were also significant for p < 0.05 except for the Upstream trend in panel (a).

25. Fig 9 to me it's not clear what the different subsurface layers mean. Soil? Groundwater? Saprolite?

We clarify in the revised manuscript that the soil layer in the figure (now Figure 10) represents unconsolidated deposits in the valley, which can be up to 1 km thick. Near the river channel, these are mostly alluvial deposits which overlay glacial deposits. Further up the slopes (but still in the valley) there are no alluvial deposits and the glacial deposits are uncovered. We now state in the figure caption:

The subsurface cross section represents the thick layer (up to 1km) of sedimentary deposits in the valley (see Sect. 2).

26. L475 Doesn't the conclusion that NDVI was leading for ET follow from a model in which ET is calculated from NDVI? And, why did NDVI increase? The authors describe that warming temperatures were a reason for NDVI to increase, but said that temperature did not influence (evapo)transpiration directly? Precipitation amounts were much lower, and an increase in storms < 1mm don't bring much moisture to the soil either, that will be evaporated directly from the plant leaves. How can the climatic conditions be more favorable, while moisture and temperature apparently don't play a direct role?

We have sought to bring clarity to this issue in a couple of ways. First, we have added a new Figure 3 (see below), which shows the causal links between temperature, land use, NDVI, PET, and ET. Our ability to differentiate the effects of temperature and land use requires assumptions about the separability of the effects of temperature and NDVI. Notably, we evaluate the hypotheses under the assumption that the effect of temperature occurs only through the effect on PET, and that the effect of land use occurs only through the effect of NDVI.

Second, in the results and discussion, we clarify how this assumption might be affected by additional factors. In particular, we evaluate how ET in different combinations of land use

(before and after 2000). We find the largest increases in ET (per pixel) occurred where land use changed to crops or mosaic (ie, orchards). However, we also find that ET increased in all land use classes where land use was consistent. In particular, this implies that intensification occurred in agricultural classes (ie, increasing fertilizer was used) and/or that temperature had an indirect effect on NDVI (eg, via earlier snowmelt and greening, which we view as the likeliest scenario in natural land use classes such as forest, and shrubs/grasses). In other words, Hypothesis 5 must be interpreted in light of the possibility that NDVI was affected not only by land use but also by temperature. While we cannot state with certainty the reasons that NDVI increased, these two hypotheses are supported by other studies that have considered these issues within the Himalayan region. In particular, Rashid et al (2015) show that climate change will lead to greater vegetation activity in high elevation regions of the Himalayas and Wetlands International South Asia (2007) describe an intensification of agriculture in the Upper Jhelum watershed.

We have clarified these points in the manuscript (L558-566):

Evapotranspiration exhibited the greatest increases in regions that transitioned to orchard cultivation, but these areas represent a small fraction of the overall watershed and increase in ET. In places where land use was unchanged, the observed increase in NDVI indicates an increase in water availability (when ET is water limited), an increase in energy availability (when ET is energy limited), or an increase in nutrient availability (where plant growth is limited by nutrient availability). In other words, such greening could arise due to agricultural intensification (via increased irrigation or fertilizer application) or increasing temperature due to climate change. These findings are consistent with other studies. For instance, agricultural intensification has been documented in the Upper Jhelum (Wetlands International South Asia, 2007) and modeling studies have demonstrated that warming temperatures will produce more favorable conditions for plant biomass growth (Rashid et al., 2015).

We also clarify the relationship among these variables by adding the following figure:

[Figure]

Figure 3: Drivers of evapotranspiration (ET). Temperature (T) modulates ET directly through the effect on potential evapotranspiration (PET), or indirectly by changing growing conditions, vegetation structure and phenology, and therefore NDVI. Land use modulates ET by directly changing land cover characteristics, or indirectly by affecting air temperature. We evaluate the effects depicted by the solid red lines and decouple the effects of the dashed lines via the Hargreaves model and maps of land use change (see text for details).

27. S1 pleas add 1:1 lines in the mass-mass plots

Thanks for the comment. We would like to note that double-mass plots should be linear, but not necessarily 1:1 (e.g., if one gauge reports more or less streamflow on average). To address the essence of your comment we now display linear trends on the double mass plots (see Figure S1 in the revised SI)

28. S2 what does the overestimation of precipitation say about the accuracy of your input data? What do the authors reckon is the importance of the spatial distribution of rain vs. the basin-average precipitation estimation. Spatial distribution might be very important, which is indicated by the small change in Q for various gauges. Commenting on this would be appreciated.

We don't have a way to validate the uncertainty on precipitation aside from using the water balance to see how far away we are from water balance closure. With that said, there is a clear and statistically significant decrease in spring precipitation at all three gauges, similar to the basin-average precipitation (which we note in the caption to Figure S2). Lastly, we reiterate that because we utilize datasets that span both periods of analysis (before and after 2000), this issue would only affect our analysis if there is a new bias that appears (or disappears) between the periods. In addition to adding Figure S5 to the SI (shown above in response to comment #2), we have added the following text to Section S2:

Lastly, we note that the annual decrease in precipitation was comparable across all three observation stations: -158 mm at Kupwara (northwestern station), -96 mm at Srinagar station (middle station), and -129 mm at Pahalgam (southeastern station). The Pahalgam station is at the highest elevation (approximately 2700 m), while the Kupwara and Srinagar stations are at lower and comparable elevations (1600 m).

29. S6 what was assumed when zero or only few (defined as less than 5 in the text) images were available?

This comment refers to the number of summer landsat observations (for each pixel) prior to 2000 that were available to estimate NDVI. In some pixels, we had less than 5 cloud-free observations. We do not assume anything for these pixels, but rather note that they may bias the results downward, thereby making our analyses conservative. In the revised SI, we provide a histogram of summer landsat observations to demonstrate that the net effect is likely small, because most pixels had a larger number of good observations. Please see our response to comment #21 for this figure.

References:

Kampf, S. K., Burges, S. J., Hammond, J. C., Bhaskar, A., Covino, T. P., Eurich, A., ... & Willi, K. (2020). The case for an open water balance: Re‑envisioning network design and data analysis for a complex, uncertain world. Water Resources Research, 56(6), e2019WR026699.

Mahmood, R., Babel, M. S., & Shaofeng, J. I. A. (2015). Assessment of temporal and spatial changes of future climate in the Jhelum river basin, Pakistan and India. Weather and Climate Extremes, 10, 40-55.

Nathan, R. J. and McMahon, T. A. (1990): Evaluation of Automated Techniques for Baseflow and Recession Analysis, Water Resources Research, 26, 1465–1473, https://doi.org/10.1029/WR026i007p01465.

---

## Referee Report (RR1)

I reviewed the author's response and revised manuscript of the Penny et al. submission. The review comments were comprehensively dealt with, and resulted in a great number of changes to the manuscript.

I was glad to see that the authors adapted the framework of the manuscript and that the multiple-hypotheses approach is now worked better into the manuscript. They added statistical tests to their work, as well as uncertainty estimates, which convinced me of the validity of their story. I also think that the comparison using two different precipitation distribution methods is a good addition, that will be of interest to future readers.

A few minor comments:

L352 Should this be related to? (instead of related with)

Fig. 1 Caption: I assume that green means no water or snow at the surface? Perhaps mention that directly in the caption.

Fig. 5 Caption: Please add the meaning of S, M and L in panel b to the caption.

Fig. 10: Why are there different types of vegetation (?), the green shapes on the hillside. It is not clear to me which different vegetation types (?) these symbols represent. Making that more explicit in the figure or caption, or homogenizing the symbols would be helpful.

Fig. S11, does 'cumec' stand for cubic meter per second? I have never seen this unit description before, and I'd suggest changing it to m$^3$/s.

---

## Author Response (AR2)

Dear Dr. Slater,

Many thanks to you and the reviewers for consideration of this manuscript. We have addressed each of the minor corrections requested by Reviewer 2 and have uploaded final versions of the manuscript, supplementary material, and figures.

Sincerely,

Gopal Penny, on behalf of all authors